# PARAMETER AVERAGING FOR FEATURE RANKING

## ABSTRACT

Neural Networks are known to be sensitive to initialisation. The methods that rely on neural networks for feature ranking are not robust since they can have variations in their ranking when the model is initialized and trained with different random seeds. In this work, we introduce a novel method based on parameter averaging to estimate accurate and robust feature importance in tabular data setting, referred as XTab. We first initialize and train multiple instances of a shallow network (referred as local masks) with *different random seeds* for a downstream task. We then obtain a global mask model by *averaging the parameters* of local masks. We show that although the parameter averaging might result in a global model with higher loss, it still leads to the discovery of the ground-truth feature importance more consistently than an individual model does. We conduct extensive experiments on a variety of synthetic and real-world data, demonstrating that the XTab can be used to obtain the global feature importance that is not sensitive to sub-optimal model initialisation.

## 1 INTRODUCTION

Neural networks (NNs) have gained wide adaption across many fields and applications. However, one of the major drawback of NNs is their sensitivity to weight initialisation (McMahan et al., 2017). This drawback is not critical for most classification and regression tasks, and is less obvious in applications such as explainability in most computer vision (CV) tasks. The problem is more obvious in settings, in which we pay attention to individual features (e.g., a feature in tabular data, or a pixel in the image) rather than group of features (e.g., a region in the image). And it becomes critical in settings, in which we might need to make costly decisions based the explanation that the model gives for its outcomes. Few such applications include disease diagnosis in clinical setting, drug repurposing in drug discovery, and sub-population discovery for clinical trials, in all of which the discovery of important features is critical. In this work, we investigate the robustness of neural networks to model initialisation in the context of feature ranking, and conduct our experiments in tabular data setting.

The methods developed to explain predictions should ideally be robust to model initialisation. This is especially important to build trust with stakeholders in fields such as healthcare. In this work, we define the "robustness" as one, in which the feature ranking from the model is not sensitive to sub-optimal model initialisation. Some examples of robust models are seen in tree-based approaches such as the random forest (Breiman, 2001) and XGBoost (Chen & Guestrin, 2016), especially when they are used together with methods such as permutation importance. In these methods, each tree is grown by splitting samples on each decision point by using an impurity metric such as Gini index for the classification task. The importance of a feature in a single tree is typically computed by how much splitting on a particular feature reduces the impurity, which is also weighted by the number of samples the node is responsible for. The importance scores of the features are then averaged across all of the trees within the model to get their final scores. It is this averaging that might be one of the reasons why these models are robust and consistent when used for feature ranking. However, we should make a distinction between the robustness of a method and the correctness of its feature ranking as tree-based methods are known to have their shortcomings (Strobl et al., 2007; Li et al., 2019; Zhou & Hooker, 2021). To get a robust explanation using neural networks, we could use an ensemble approach by training multiple neural network-based models to get feature importance, and use the majority rule to rank them. However, the ranking of features by using the ensemble of models may still not be easy in cases where the same feature(s) get ranked equally likely across different positions by the models. Moreover, the ensemble approach requires us to store all models so that we can use them to explain a prediction at test time, which is not ideal. Instead, in this work, we propose a novel

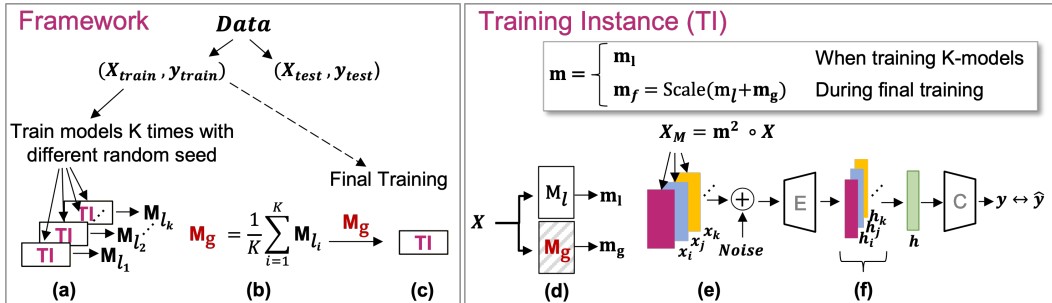

Figure 1: **Left:** Framework; **a)** Train the models K-times with different random seeds, **b)** Obtain the global mask, **c)** Final training by using global mask (frozen weights) and a new local mask (trained). **Right:** Details of each training instance; **d)** Generating mask from input, **e)** Feature bagging using masked input, **f)** Aggregating the embeddings of the subsets. **E:** Encoder, **M:** Mask, **C:** Classifier.

method, in which we obtain a single global mask model that is based on *averaging the parameters of multiple instances (local masks) of the same model*. We take advantage of the sensitivity of NNs to initialization by initializing and training each local mask *with a different random seed*. We show that although the global model might have a higher loss than an individual model, it ranks features more correctly and consistently, and hence can be used to extract the feature importance.

Our primary contributions in this work are the following: We obtain a global model by averaging the parameters of multiple instances of a *shallow* neural network trained *with different random initialisation* and use it to extract feature importance. The global model obtained in this manner might have a higher loss than any of the individual models (McMahan et al., 2017). We show that although this is true, the global model is still able to discover the ground-truth feature importance more consistently than an individual model does. We also demonstrate that weight regularization such as dropout and weight-clipping can improve the robustness and consistency of the global model. We show that the existing the state of the art (SOTA) methods proposed for feature ranking or selection are not robust to model initialisation. Finally, we provide insights via extensive empirical study of parameter averaging using both synthetic and real tabular datasets.

## 2 METHOD

Parameter averaging is extensively studied in the context of Federated Learning (McMahan et al., 2017), in which individual models are trained on datasets stored in different devices, and a global model is obtained by averaging individual models in various ways. For example, the naive parameter averaging is shown to give a lower loss on full training set than any individual model trained on a different subset of the data *when the individual models are initialized with same random seed* (McMahan et al., 2017). It is well known that the loss surface for typical neural networks is non-convex (McMahan et al., 2017) and, hence, averaging parameters of models could result in a sub-optimal global model, especially when their parameters are initialised differently. However, the loss surfaces of over-parameterized NNs are shown to be well behaved and less prone to bad local minima in practice (Choromanska et al., 2015; Dauphin et al., 2014; Goodfellow et al., 2014). In light of these observations, we investigate settings, in which we can combine multiple models *that are initialized and trained with different random seeds* to obtain a global model that is less sensitive to sub-optimal initialisation of any individual model. So, in this work, we propose a framework to obtain such a global model that can be used for both feature ranking and selection. We show that global model is able to extract feature importance correctly and consistently especially *when the network architecture is shallow*. We also show that this behaviour breaks down for deep architectures although regularizing their weights still helps improve them.

### 2.1 TRAINING

Figure 1 shows our framework, in which we use a shallow neural network as mask generator that in turn is used to learn important features and their weights for a downstream task. The hidden layer

of the mask generator has the same number of hidden units as the number of input features. We use a sigmoid activation at its output to have values in $[0, 1]$ range. In this work, without the loss of generality, we use the classification task for the experiments as shown in Figure 1 (right). We also adopt the SubTab framework (Ucar et al., 2021), in which we use a shallow, overcomplete encoder architecture followed by a classifier and train them together with the mask generator on subsets of the features. Please note that one can train the mask generator with a simple classifier in a standard way instead of integrating it into the SubTab framework. We conduct an ablation study to show that our method is agnostic to whether one uses SubTab or not in Section E of the Appendix. The summary of model architectures and hyper-parameters is in Section C.1 and Table A1 in the Appendix.

**High-level overview:** A mask generator, an encoder and a classifier are trained $K$ times using the same training set. $K$ training runs can be parallelized in a distributed setting, or can be run in series on the same machine. At the beginning of each run, we change the random seed before initialising all models (i.e. mask, encoder, and classifier) using Kaiming He uniform initialization (He et al., 2015) with the gain of $\sqrt{5}$ for linear layers. At the end of each training run, we keep the learned weights of the mask model referred as the *local* mask. So we have $K$ different set of weights for the same mask model at the end of $K$ runs. Then, we average the parameters of $K$ local mask models to obtain the weights of the global mask model. In Section 3.4, we show that the global mask is good at extracting feature importance, but it can be sub-optimal for the classification task since it has a higher loss than an individual model as shown in Figure 6. Thus, we initialise and train the models one final time, during which we combine the output of global mask model (weights frozen) with the one from a local mask (trained). The local mask is trained to gain back any potential loss in classification performance. We should note that one can also choose to fine-tune the global mask, but we prefer to use it as a reference to improve on in the final training.

**Training to obtain a local mask:** We train a local mask generator, a classifier and an encoder for a downstream task. Mask generator, $M_l$, gets data $X$, and generates a mask $m = m_l$. We then mask the input $X$ by using an entry-wise multiplication with $m^2$ to generate a masked input $X_M$. We use $m^2$ instead of $m$ to push low values in $m$ towards zero. In our experiments, we observed that using $m^2$ works better than $m$.

$$m = M_l(X), \text{ and } X_M = m^2 \odot X \tag{1}$$

Inspired by the proposal for subsetting features in SubTab (Ucar et al., 2021), we then generate subsets of data by dividing the features of $X_M$: $\{x_i, x_j, x_k, \dots\}$. Learning from subsets of features is shown to be effective in learning good representations for downstream tasks such as classification while enabling parameter sharing between the features of the tabular data (Ucar et al., 2021). We also add noise to randomly selected features in each subset since we observe that adding noise improves classification performance and the robustness of feature ranking as discussed in Section L of the Appendix. To add noise, we first generate a binomial mask, $\beta$, and a noise matrix, $\epsilon$, both of which have the same shape as the subsets, and are re-sampled for each subset. The entries of the mask are assigned to 1 with probability $p$, and to 0 otherwise. As an example, the corrupted version, $x_{ic}$ of subset $x_i$ is generated as following:

$$x_{ic} = (1 - \beta) \odot x_i + \beta \odot \epsilon, \qquad \text{where } i \in \{1, 2, \dots\}. \tag{2}$$

Please note that different noise types can be used to generate $\epsilon$. In this paper, we mainly experiment with Gaussian noise, $\mathcal{N}(0, \sigma^2)$, except for SynRank100 dataset, for which we use swap noise (Ucar et al., 2021). The encoder takes each of the corrupted subsets $\{x_{ic}, x_{jc}, x_{kc}, \dots\}$, and projects them up to generate corresponding embeddings, $\{h_i, h_j, h_k, \dots\}$. As in SubTab (Ucar et al., 2021), we aggregate the embeddings by using mean aggregation to get the joint embedding, $h$, as shown in Figure 1. Finally, the classifier makes a prediction using the joint embedding $h$. We minimize the total loss by using the objective function in Equation 3 that consists of two loss functions; i) Cross entropy for the classification task (Equation 4), ii) Mask loss consisting of Gini index and an extra term taking the mean over the entries of the generated mask to induce sparsity (Equation 5):

$$\mathcal{L}_{total} = \mathcal{L}_{task} + \mathcal{L}_{mask} \tag{3}$$

$$\mathcal{L}_{task} = \frac{1}{N} \sum_{i=1}^{N} -y_i * log(\hat{y}_i) - (1 - y_i) * log(1 - \hat{y}_i) \tag{4}$$

$$\mathcal{L}_{mask} = \frac{1}{N} \sum_{i=1}^{N} \frac{1}{D} \sum_{j=1}^{D} g_{ij} + \frac{1}{N} \sum_{i=1}^{N} \frac{1}{D} \sum_{j=1}^{D} m_{ij}, \quad \text{where } g = 1 - m^2 - (1 - m)^2 \tag{5}$$

We update the parameters of the local mask, encoder, and classifier using the total loss (Equation 3). At the end of each $k^{th}$ training run, we collect the parameters of the local mask, $M_{l_k}$.

**Final training:** Once the $K$ training runs are completed, we obtain a global mask model by averaging the parameters of $K$ individual local masks as shown in Equation 6:

$$\boldsymbol{M_g} = \frac{1}{K} \sum_{k=1}^{K} \boldsymbol{M_{l_k}} \tag{6}$$

where $\boldsymbol{M_{l_k}}$ is the local mask generator collected at $k^{th}$ run, and $\boldsymbol{M_g}$ is the global mask. $\boldsymbol{M_g}$ might give sub-optimal performance in downstream task since it is shown to result in higher loss than a local mask model (Section 3.4). But, we do not want to lose the benefits that come with averaging the parameters in $\boldsymbol{M_g}$ by fine-tuning it. So, to avoid the potential degradation in performance, we do one final training. We change the random seed, and initialize all the models. In this step, we don't train the global mask, but rather train a new local mask together with the encoder and classifier as before. However, the difference from the previous training instances is that the mask used for masking the input data is obtained by summing the output of global mask (with frozen weights) and local mask (being trained), followed by scaling this output to make sure that the maximum entry in the mask is 1 as shown in Equations 7, and 8:

$$\boldsymbol{m_g} = \boldsymbol{M_g(X)} \text{ and } \boldsymbol{m_l} = \boldsymbol{M_l(X)} \tag{7}$$
$$\boldsymbol{m_f} = (\boldsymbol{m_g} + \boldsymbol{m_l})/C \text{ where } C = max(\boldsymbol{m_g} + \boldsymbol{m_l}) \text{ and } \boldsymbol{X_M} = \boldsymbol{m_f}^2 \odot \boldsymbol{X}. \tag{8}$$

We should note that $C$ is a scaler, i.e. maximum entry in $\boldsymbol{m_g} + \boldsymbol{m_l}$ sum. We use the same loss functions described in equations 3, 4 and 5 to update the parameters of the local mask, encoder and classifier. We should note that $\boldsymbol{m_f}$ can be computed in various ways such as using a gating mechanism similar to input and forget gates in LSTMs (Hochreiter & Schmidhuber, 1997). We can also choose to keep updating $\boldsymbol{M_g}$ in a sequential manner rather than averaging parameters of multiple models all at once. We leave these ideas as future work. Our method is summarized in the Algorithms 1 and 2 in the Appendix.

## 2.2 TEST TIME

At test time, we use $\boldsymbol{m_g}$ shown in Equations 7 to infer the feature importance. $\boldsymbol{m_g}$ is shown to give a robust global ranking of features in our experiments. In XTab, the importance score for a feature is the mask weight in the final generated mask. The mask weight indicates the feature's relative importance, and we rank the features based on their mask weights. We extract the global feature importances for test set by getting mask values for all samples and computing the mean values over the samples for each feature:

$$\hat{\boldsymbol{f_i}} = \boldsymbol{M_g(x_i)} \text{ and } \hat{\boldsymbol{F}} = \frac{1}{N} \sum_{i=1}^{N} \hat{\boldsymbol{f_i}}, \tag{9}$$

where $\hat{\boldsymbol{f_i}} \in R^d$ represents mask weights (i.e. feature importance) for the $d$ number of features in sample $\boldsymbol{x_i} \in R^d$, hence $\hat{\boldsymbol{f_i}}$ gives an instance-wise feature importance for $i^{th}$ sample. $\hat{\boldsymbol{F}} \in R^d$ gives the mean of mask weights over N samples and we use it when computing the global feature importance. Please note that "global" is an overloaded term in the sense that "global" refers to a feature's overall importance across all samples in a dataset in the context of feature ranking i.e. it is not related to the term "global" in the global mask model. Finally, when ranking the categorical features, we can rank individual one-hot encoded features to show importance of each sub-category. We can also sum the weights of each one-hot encoded feature to get the overall weight for the parent category. We use both when comparing our method to other methods in Sections L.4 and L.5 of the Appendix.

## 3 EXPERIMENTS

We conduct extensive experiments on diverse set of tabular datasets including six syntetic datasets as well as real world datasets such as UCI Adult Income (Income) (Kohavi, 1996), and UCI BlogFeedback (Blog) (Buza, 2014). We conduct our initial experiments on synthetic datasets since their ground-truth important features are known. We also compare global feature rankings obtained by

the proposed method for synthetic datasets to those given by recently published neural network-based methods such as Invase (Yoon et al., 2018), L2X (Chen et al., 2018), TabNet (Arık & Pfister, 2021), and Saliency Maps (Simonyan et al., 2013). We report the detailed results on Income and Blog datasets in Sections L and M while additional experiments using synthetic datasets from L2X (Chen et al., 2018) are in Sections J of the Appendix respectively. For completeness, we also make comparisons with methods such as permutation feature importance used together with random forest (Breiman, 2001) and gradient boosting classifier (Pedregosa et al., 2011) in Section F of the Appendix.

## 3.1 DATA

**SynRank dataset:** We generate a synthetic dataset, referred as SynRank, consisting of training and test sets with 10k samples each for a binary classification to evaluate whether our method can rank important features in correct order. We first generate data $X$ from 10-dimensional standard Gaussian with no correlations across the features $\mathcal{N}(\mathbf{0}, \mathbf{I})$. We then shift the sixth feature , $f_6$, to be centered around $-10$ for the first $45\%$ of samples. For the next $35\%$ of the samples, we shift the first feature $f_1$ to be centered around 10. The remaining $20\%$ of the samples are kept same as is. We generate the label $Y$ by sampling it as a Bernoulli random variable with $P(Y = 1|X) = 1/(1 + g(X))$. In this case, $g(X)$ is defined as $exp(f_6)$, $exp(f_1)$ and $exp(f_2)$ for the $45\%$, $35\%$ and $20\%$ of the samples respectively. So the first $45\%$ and $35\%$ of the samples will be labeled as 1 and 0 with a high probability, respectively. For the remaining $20\%$ samples, we can expect the proportions of class labels to be similar since $f_2$ is from a standard Gaussian with $\mu = 0$. Based on this dataset, we expect that our method discovers the global feature importance ranking as $f_6 > f_1 > f_2$.

**SynRank100 dataset:** This dataset is same as the SynRank, but it has 100 features instead of 10. The features $f_{100}$, $f_1$, and $f_{75}$ are the equivalents of $f_6$, $f_1$, $f_2$ in SynRank respectively, and hence the feature ranking is $f_{100} > f_1 > f_{75}$ while the remaining 97 features are uninformative.

**Synthetic datasets from L2X:** We run experiments on four synthetic datasets used for binary classification in L2X (Chen et al., 2018). For each dataset, we have 10k training and 10k test set. In first three datasets, we generate data $X$ from 10-dimensional standard Gaussian and assign labels using $P(Y = 1|X) = 1/(1 + g(X))$ in each dataset, where $g(X)$ is defined in the following way: **i)** *XOR*: $exp(f_1 * f_2)$, **ii)** *Orange Skin*: $exp(\sum_{i=1}^{4} f_i^2 - 4)$, and **iii)** *Nonlinear Additive*: $exp(-100 * sin(2 * f_1) + 2 * |f_2| + f_3 + exp(-f_4))$. In the fourth dataset, **iv)** *Switch*: We generate $f_{10}$ from a mixture of two Gaussians centered at $\pm 3$ respectively with equal probability. If $f_{10}$ is from the $\mathcal{N}(3, 1)$, then we use $\{f_1, f_2, f_3, f_4\}$ to generate Y from the Orange Skin model. Otherwise, we use $\{f_5, f_6, f_7, f_8\}$ to generate Y from the Nonlinear Additive model. $f_9$ is not used when generating labels. We include some results for L2X Switch in the main paper while the results on other datasets can be found in Section H, I and J of the Appendix.

**Income:** Income is a public dataset based on the 1994 Census database (Kohavi, 1996). It is used for a classification task of predicting whether the income of a person exceeds \$50K/yr by using heterogeneous features such as age, gender, education level and so on. It contains 32.5k and 16k samples for training and test sets respectively. The dataset has 14 attributes consisting of 8 categorical and 6 continuous features. We dropped the rows with missing values, and encoded categorical features using one-hot encoding. Once we encode the categorical features as one-hot, we end up with 105 features in total.

**BlogFeedback:** Referred as Blog in this work, it is a UCI dataset (Dua & Graff, 2017) and contains the number of comments in the upcoming 24 hours for blog posts. It includes 281 variables consisting of 280 integer and real valued features and 1 target variable indicating the number of comments a blog post received in the next 24 hours relative to the basetime. We converted the target to a binary variable to use the data for a classification task of predicting whether there is a comment for a post.

The more details on Income and Blog datasets are added in Section B of the Appendix.

## 3.2 COMPARING GLOBAL MODEL TO LOCAL MODELS

We start our experiments with the classification task on SynRank dataset to get insights into how parameter averaging works for extracting feature importance[1] as shown in Figure 2. The results for

---

[1]Unless specified otherwise, when we say feature importance, we refer to global feature importance.

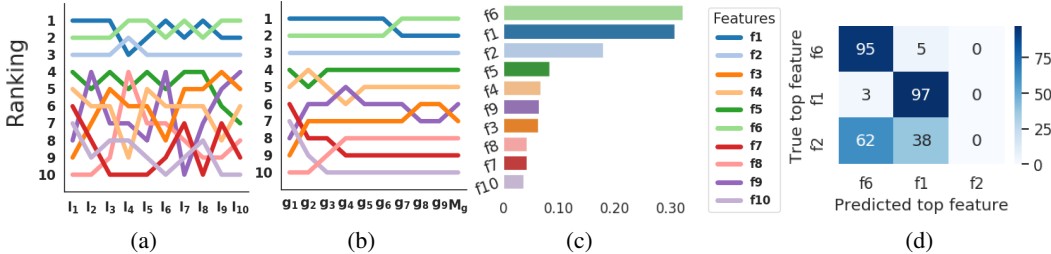

(a)        (b)        (c)        (d)

Figure 2: **SynRank dataset: a)** Feature rankings from each of 10 local masks $M_{l_k}$, referred as $l_k$ in the figure, obtained at a particular training run for 10 separate runs. **b)** Feature rankings from the global model, obtained by averaging the parameters of individual models up to a specific run i.e. cumulative average (CA). For example, $g_3$ corresponds to the global model obtained by averaging the parameters of first 3 local masks ($l_1, l_2, l_3$). **c)** The feature importance weights from $M_g$ obtained by averaging the parameters of all local masks. **d)** Instance-wise feature importance: Comparing ground-truth top feature to predicted top feature from $M_g$ for each sample (shown in percentages).

L2X datasets (Chen et al., 2018) can be found in Section H while the details on hyperparameters such as $p$ used for generating the binomial mask, $\beta$, and the variance of Gaussian noise, $\epsilon \sim \mathcal{N}(0, \sigma^2)$, can be found in Section C.1 of the Appendix. We train our models on the whole training set for the downstream task 10 times, each time with *a different random seed*. We store the parameters of the trained masks, referred as local masks, from each training and denote them as $\{M_{l_1}, M_{l_2}, \ldots, M_{l_{10}}\}$. We examine the feature importance obtained from each of 10 local masks for the test set (Figure 2a). We observe that each local mask gives a slightly different ranking. More specifically, we can have different ranking, depending on which seed is used when training the models. The main reason for this variation is the model initialisation since everything else are kept same across different trainings. We then evaluate the effect of averaging over the parameters of the local masks on feature ranking in a progressive way. To do this, we obtain a global mask $M_{g_k}$ as a cumulative average (CA) over the $k$ local masks, i.e. $M_{g_k} = 1/k \sum_{i=1}^{k} M_{l_k}$ as shown in Figure 2b. For example, $M_{g_3}$ corresponds to averaging the parameters of the first three local masks i.e., $M_{l_1}, M_{l_2}, M_{l_3}$ shown in Figure 2a. We refer to $M_{g_{10}}$ as $M_g$ for simplicity in the rest of the paper. We can see that the feature ranking becomes more stable as we use more local masks in the parameter averaging to obtain the global mask model (Figure 2b).

### 3.3 EVALUATING THE CORRECTNESS OF FEATURE RANKING

Figure 2c shows the feature weights obtained from $M_g = M_{g_{10}}$. We first note that the weights of the features are correlated with the frequency and position of their ranks across all local masks. Specifically, $f_6$ is ranked a little higher on average than $f_1$ is (since $f_1$ occasionally takes $3^{rd}$ rank). Hence, $M_g$ correctly gives $f_6$ a little more weight than $f_1$ and suggests $f_2$ as the $3^{rd}$ most important feature as shown in Figure 2c. Moreover, we investigate the instance-wise feature importance for XTab. Since we optimize our models using a global objective function (Equation 3), we expect $M_g$ to be biased towards globally important features when estimating the feature importance for an individual sample as confirmed in Figure 2d. We see that $M_g$ is biased towards the globally important features, and fails to rank $f_2$ as the most important feature for 20% of the samples in the test set, for which we generate labels using $P(Y = 1 | X) = 1/(1 + exp(f_2))$ and $f_2$ is sampled from $\mathcal{N}(0, 1)$.

### 3.4 THE ROBUSTNESS AND CONSISTENCY OF PARAMETER AVERAGING

We compare the robustness and consistency of the global feature importance extracted from various methods by running each method 10 times with different random seeds on SynRank and L2X Switch datasets in Figure 3. XTab discovers the top three features as "$f_6 > f_1 > f_2$" for SynRank consistently while the rankings from TabNet, Invase, L2X and Saliency Maps are not robust to initialisation (Figure 3a). For example, TabNet has a wide variation in its rankings whereas L2X gets the ranking completely wrong. Please note that Invase and L2X are originally proposed as feature selection methods. So, we use the feature-selection probabilities for Invase and the frequency of a

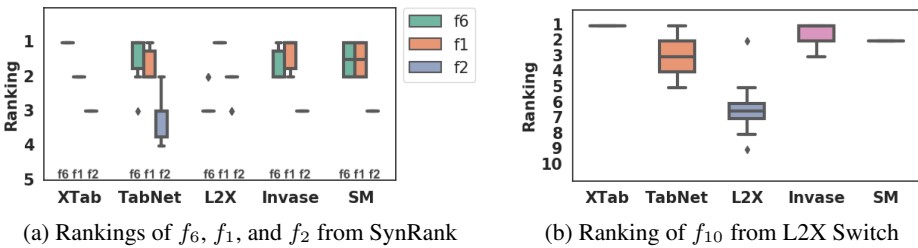

(a) Rankings of $f_6$, $f_1$, and $f_2$ from SynRank    (b) Ranking of $f_{10}$ from L2X Switch

Figure 3: Comparing different models for the consistency and correctness of their feature rankings by using SynRank and L2X Switch datasets. $f_6 > f_1 > f_2$ is the correct feature ranking for SynRank while $f_{10}$ is the most important feature for L2X Switch. Abbreviations; **SM:** Saliency Maps.

feature being selected for L2X when computing the feature rankings. Moreover, in Figure 3b, we repeat the same experiment for L2X Switch dataset and show the ranking of the most important feature $f_{10}$ by the same models. XTab ranks $f_{10}$ as the most important feature consistently while others fail. Please note that this is the hardest synthetic dataset in a way that the effect of $f_{10}$ on the sample labels is not direct, rather it influences the labels indirectly by deciding which features to be used for label generation. The details of the training for the models as well as the ranking results for all L2X datasets are in Sections C and J of the Appendix, respectively.

**The effect of weight regularization on the robustness.** We run additional experiments on SynRank under three conditions: We apply i) No weight regularization to the weights of the mask model i.e., our original setting so far, ii) Dropout with $p = 0.2$ for the layers with leaky ReLU activation, iii) Weight-clipping ($[-0.2, 0.2]$) to limit the magnitude of the weights in each layer. For each of the three cases, we train 20 separate models and compare two different settings. In the first setting, we compute the variation in the feature rankings given by 20 local mask models (top row in Figure 4a-c). In the second setting, we obtain 100 global models, each of which is obtained by averaging the parameters of 10 local models *bootstrapped* from 20 models. We compare the variation in feature rankings given by global models (second setting – bottom row in Figure 4a-c) to that of 20 local models (first setting – top row). We observe that: i) Regularization methods such as dropout and weight-clipping has a little effect in improving the variation across 20 local models (e.g., comparing $f_1$, $f_2$ and $f_6$ across three cases at the top row). ii) They do not improve the robustness of the parameter averaging significantly (e.g., comparing same features across three cases at the bottom row). iii) Parameter averaging results in more robust estimation of feature rankings (comparing top and bottom rows). Overall, the global models are able to discover important features in the correct order (e.g., $f6 > f1 > f2$), and the variation in feature ranking across global models is small (almost zero) for ground truth important features (e.g., $f_1$, $f_2$ and $f_6$ at the bottom row in Figure 4c). We should note that we conduct the same experiment for SynRank100 (Figure 4d), L2X Switch, Income, and Blog datasets, for the latter three of which the weight regularization helps improve robustness of parameter averaging (Figures 5, A8, A18, and A22 respectively). Thus, the weight regularization can help improve the robustness and weight-clipping works better than dropout in our experiments. We also observe that the parameter averaging itself pushes the magnitude of the weights towards a tight range around zero as shown in Figure A9 (Section I.1 of the Appendix), indicating a potential relationship between robustness and a tighter weight distribution.

**Exploring the mask generator with deeper architecture.** We re-run the robustness experiments by replacing the shallow mask model with a deeper model (5 hidden layers) and show that the weight regularization also helps with the robustness of parameter averaging in deeper networks although the parameter averaging does not work as well as the shallow networks especially if weight regularization is not used (see Figure A16(f) for Income dataset in the Appendix). Additional results for SynRank, L2X Switch, and Income can be found in Section F.2, I.2 and L.3 of the Appendix respectively.

**Comparing the loss and solution space of local and global models.** We consider two sets of parameters: $\theta_l$ for a local model and $\theta_g$ for the global model. We can compare the possible loss and solution space by interpolating from local to the global model: $\theta^* = (1 - \alpha) * \theta_l + \alpha * \theta_g$, where $\alpha$ is swept from 0 to 1 in 50 steps. Figure 6a-b shows two separate examples of such interpolation done using SynRank dataset. In Figure 6a, the local model ($\alpha = 0$) has the wrong feature ranking of $f1 > f6 > f2$. As we move from local model ($\alpha = 0$) to the global model ($\alpha = 1$), the estimate

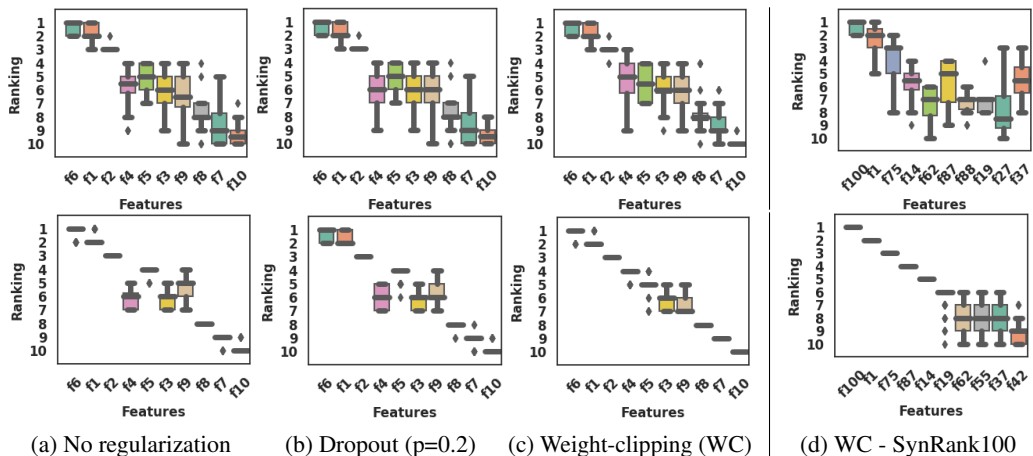

|                   |                    |                           |                        |
| ----------------- | ------------------ | ------------------------- | ---------------------- |
| (a) No regularization | (b) Dropout (p=0.2) | (c) Weight-clipping (WC) | (d) WC - SynRank100 |

Figure 4: **SynRank and SynRank100 datasets:** The important features are $f_6 > f_1 > f_2$ and $f_{100} > f_1 > f_{75}$ for SynRank and SynRank100 respectively while the rest of the features is not informative. **Top row:** Variations in feature rankings across 20 local mask models when we apply **a)** no weight regularization, **b)** Dropout ($p = 0.2$), and **c)** Weight-clipping ($[-0.2, 0.2]$) for SynRank dataset. **Bottom row**: Variations in feature rankings in 100 global models, each of which is obtained by averaging the parameters of 10 models bootstrapped from 20 local models for the same three cases. **d)** Same as **(c)** i.e. weight-clipping (WC), but the experiment is repeated for SynRank100 dataset.

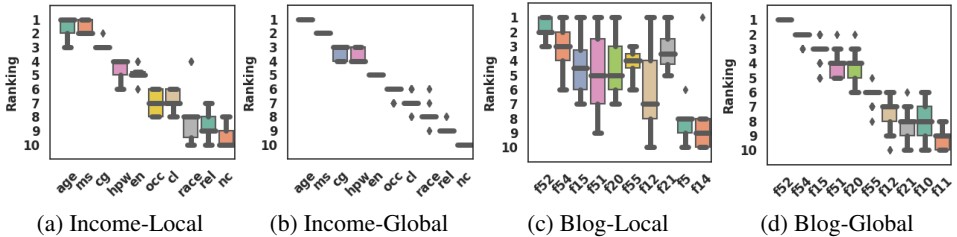

|                 |                  |              |             |
| --------------- | ---------------- | ------------ | ----------- |
| (a) Income-Local | (b) Income-Global | (c) Blog-Local | (d) Blog-Global |

Figure 5: **Income and Blog datasets:** Repeat of the experiment in Figure 4c, showing variations in feature rankings across 20 local mask models in **(a)** and **(c)** as well as variations across 100 global models in **(b)** and **(d)** (Please see Section L.3 and M.1 for more results). Abbreviations used for features in Income are; **ms:** marital status, **en:** education-num, **cg:** capital-gain, **hpw:** hours-per-week, **occ:** occupation, **rel:** relationship, **cl:** capital-loss, **nc:** native-country.

of feature ranking gets better. The global model estimates the ranking correctly ($f6 > f1 > f2$). Similarly, in Figure 6b, a different local model has the wrong feature ranking of $f6 > f2 > f1$ while the global model again gives the correct ranking. Moreover, for SynRank, the expected global feature importance weights are $\boldsymbol{F} = [\boldsymbol{F_1}, \boldsymbol{F_2}, ..., \boldsymbol{F_6}, ..., \boldsymbol{F_{10}}] = [0.35, 0.2, 0, ..., 0.45, ..., 0]$. So, we can compute the loss of the mask model by using mean squared error between expected weights and the model's estimate: $\mathcal{L} = \frac{1}{10} \sum_{d=1}^{10} (\boldsymbol{F_d} - \hat{\boldsymbol{F}}_{\boldsymbol{d}})^2$, where $\hat{\boldsymbol{F}}$ is defined in Equation 9. Thus, we plot how this loss changes as we interpolate from local to global model in Figure 6c, which corresponds to the interpolations in Figure 6a-b. It indicates that the loss increases as we move towards the global model, which is mainly due to the fact that the noise floor (**nf**) increases in both cases as shown in Figure 6a-b and that the weight of $f6$ moves away from 0.45 in the case of Figure 6b. Averaging parameters of the multiple models with different initialisation is known to give a global model that might have a higher loss than any of the local models (McMahan et al., 2017). However, we show that the global model is still better at estimating feature ranking and more robust than the local models.

# 4 RELATED WORKS

**Parameter averaging** Averaging parameters to get a global model has been extensively studied in the Federated Learning setting under different assumptions; i) The convex optimisation under IID

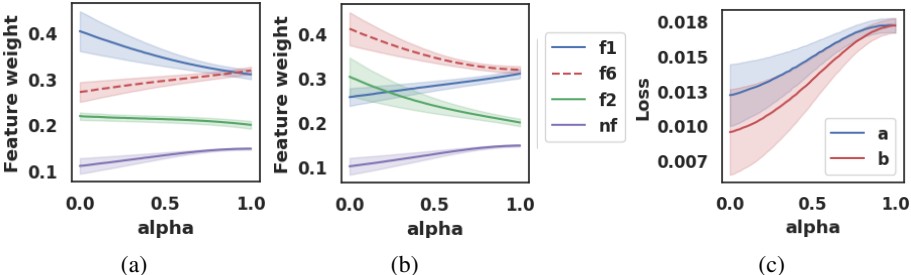

Figure 6: **SynRank: (a-b)**: Two examples showing the changes in the weights of features estimated by a model that is obtained by interpolating between a single local model and the global model: $\theta^* = (1 - \alpha) * \theta_l + \alpha * \theta_g$. As we move from local model ($\alpha = 0$) to the global model ($\alpha = 1$), the estimate of feature ranking gets better (i.e., $f6 > f1 > f2$) even though the loss of the model gets worse as shown in **(c)**. **nf:** Noise floor, referring to average weight of the rest of the uninformative features i.e., $f3 - f5, f7 - f10$. The plots show the mean and 95% confidence intervals computed by repeating the experiments with 100 global models, each of which is obtained by bootstrapping 10 out of 20 local models. In (a-b), global models are interpolated with two different local models.

data assumption, in which it is shown that the global model is no better than a single model in the worst-case (Arjevani & Shamir, 2015; Zinkevich et al., 2010; Zhang et al., 2012). ii) The non-convex optimisation under IID and non-IID data assumptions, in which individual models are initialized from *the same random initialization* to avoid bad local minima before training each independently (McMahan et al., 2017). Parameter averaging using models with same initialisation is studied under different contexts as well (Wortsman et al., 2022; Polyak & Juditsky, 1992; Izmailov et al., 2018). In Wortsman et al. (2022), the authors average the parameters of multiple models, each of which is obtained by fine-tuning a pre-trained model by using different hyper-parameters. In this case, the fine-tuning process starts from the same initial model i.e., pre-trained model. The parameter averaging in (Polyak & Juditsky, 1992; Izmailov et al., 2018) is done by averaging the parameters of the same model along the trajectory of stochastic optimisation during the training. Moreover, dropout method is previously shown to approximate model averaging implicitly (Srivastava et al., 2014; Goodfellow et al., 2013). However, we show that the dropout alone is not enough to achieve robustness in Section 3.4. In this work, we study non-convex setting under the IID assumption, and consider averaging model parameters obtained across multiple training runs to produce the final global model. What differentiates our method from aforementioned works is that we initialise the models with *different random seeds*. Although averaging the parameters of the models trained with *different random seeds* is shown to lead to a bad local minima (McMahan et al., 2017), we show that the global model obtained in this way gives a robust estimate of the feature importance and can be used for feature ranking. We review other related works in Section D of the Appendix.

## 5 CONCLUSION

In this work, we show that a global model obtained by averaging the parameters of multiple instances of a shallow network trained with different random seeds can be used to estimate global feature importance and that its estimates are not sensitive to sub-optimal initialisation of individual models. Furthermore, regularization methods can enhance the robustness of parameter averaging. We give insights into how parameter averaging can be useful for feature ranking through extensive experiments using synthetic and real tabular datasets. Our method can also be extended to other modalities such as images, graph etc. and we leave it as a future work. Finally, the following are some of the shortcomings of our approach; i) The global model is biased towards globally important features and hence instance-wise feature importance will be biased, ii) We still need to do hyper-parameter search for feature bagging, noise etc., iii) The features in the real world datasets can have more intricate relationships such as multicollinearity, making the ranking of the features difficult, in which case our method can be used for feature selection rather than feature ranking, and iv) Our method needs additional compute during training, but this can be eliminated by integrating our method into $K$-fold cross validation, assuming $K \geq 10$.

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

## A  ALGORITHM

---

**Algorithm 1:** Main learning algorithm

---

**input:** batch size N, the structure of encoder (e), classifier (c), mask (m);

```
# Set the seed and do train-test split
```
$seed = 57$;
```
# Training and test sets.
```
$X_{train}, X_{test} = get\_data(random\_seed = seed)$;
```
# Initialize a list to hold mask models from each of K-training
instances.
```
$masks\_list = []$;

```
# Train models K-times, each time with a different random seed.
```
**for** $i_k$ *in* $\{K\}$ **do**

    ```# Change random seed and initialize models```
    $seed = seed + 17$;
    initialize all models (e, c, m) with new seed;
    ```# Train mask, classifier & encoder and return trained mask```
    ```model.```
    $\_, \_, M_{l_k}, \_ = train\_for\_downstream(training = (X_{train}, y_{train}))$
    ```# Collect parameters of the local mask model.```
    $masks\_list.append(M_{l_k})$;

**end**

```
# Average the parameters of the masks collected to obtain
global mask
```
$M_g = \frac{1}{K} \sum_{k=1}^{K} M_{l_k}$;
```
# Change the random seed and initialize models
```
$seed = seed + 17$;
initialize all models (e, c, m) with new seed;
```
# Train mask, classifer and encoder while using global mask
(frozen)
```
$\boldsymbol{e}, \boldsymbol{c}, \boldsymbol{M_l}, \boldsymbol{M_g} = train\_for\_downstream(training = (X_{train}, y_{train}), M_g = M_g)$;
```
# Return Encoder (e), Classifier (c), Local and Global masks
```
**return** $\boldsymbol{e}, \boldsymbol{c}, \boldsymbol{M_l}, \boldsymbol{M_g}$ ;

---

---

**Algorithm 2:** Training routine used for the downstream task

---

**def train_for_downstream**(training, val=None, $\boldsymbol{M_g}$=None)

   $X_{train}, y_{train} = training$

   **foreach** *epoch* $e \in Epochs$ **do**

      **foreach** *batch* $(X_b, y_b) \in (X_{train}, y_{train})$ **do**

         `# Get the output from the local mask`

         $m = M_l(X_b)$

         `# If the global mask is provided, get its output`

         **if** $M_g \neq None$ : **then**

            with no gradient:

               $m_g = M_g(X_b).detach()$

            `# Final mask is the sum of local and global masks,`

            `scaled by the max entry in the sum.`

            $m = (m + m_g)/max(m + m_g)$

         `# Get the masked input using the square of the mask`

         $X_M = m^2 \odot X_b$

         `# Generate subsets of data (i.e.  feature bagging)`

         $x_i, x_j, x_k, ... = generate\_subsets(X_M)$

         `# Obtain embeddings`

         $h_i, h_j, h_k, ... = [encoder(x) \quad for \quad x \quad in \quad [x_i, x_j, x_k, ...]]$

         `# Aggregate embeddings to get the joint embedding`

         $h = aggregate(h_i, h_j, h_k, ...)$

         `# Get predictions using the joint embedding`

         $y_{pred} = classifier(h)$

         `# Compute losses`

         $L_{task} = CrossEntropy(y_{pred}, y_b)$

         $L_{mask} = Gini(m)$

         $L = L_{task} + L_{mask}$

         `# Update models.  Note that the global mask is not`

         `trained.`

         backprop and update **local mask**, **encoder** and **classifier**

      `# If validation set is provided, run validation using the`

      `steps above (no loss computation).`

   `# Return Encoder, Classifier, Local and Global masks`

   **return** $encoder, classifier, \boldsymbol{M_l}, \boldsymbol{M_g}$

---

# B  DATA

## B.1  ADULT INCOME DATASET

Adult Income (Income) is a public dataset based on the 1994 Census database (Kohavi, 1996). It is used for a classification task of predicting whether the income of a person exceeds $50K/yr by using heterogeneous features such as age, gender, education level and so on. It contains  32.5k and  16k samples for training and test sets respectively.

**Train-Validation-Test Split:** Training and test sets are provided separately (Kohavi, 1996). We split the training set into training and validation sets using 80-20% split to search for hyper-parameters. Once hyper-parameters was fixed, we trained the model on the whole training set.

**Features:** The dataset has 14 attributes consisting of 8 categorical and 6 continuous features. We dropped the rows with missing values, and encoded categorical features using one-hot encoding. Once we encode the categorical features as one-hot, we end up with 105 features in total. Features are normalized by subtracting the mean and dividing by the standard deviation, both of which are computed using training set.

**Class imbalance:** It is an imbalanced dataset, with only 25% of the samples being positive.

## B.2  UCI BLOGFEEDBACK DATASET

Referred as Blog in this work, it contains the number of comments in the upcoming 24 hours for blog posts. Although the dataset can be used for regression, we turn it to a binary classification task to predict whether there is a comment for a post or not.

**Train-Validation-Test Split:** UCI (Dua & Graff, 2017) provides one training set, and 60 small test sets. We combined all the test sets into one test set. We split training set to training and validation using 80-20% split to search for hyper-parameters. We trained the final model using all of the training set.

**Features:** It includes 281 variables consisting of 280 integer and real valued features and 1 target variable indicating the number of comments a blog post received in the next 24 hours relative to the basetime. We converted the target (the last column in the dataset) to a binary variable, in which 0/1 indicates whether the blog post received any comments. Similarly to Income dataset, we used standard scaling to normalize the features.

**Class imbalance:** $\sim 36\%$ of the samples are positive in training set while it is $\sim 30\%$ in the test set.

## B.3  SYNTHETIC DATASETS FROM L2X:

We run experiments on four synthetic datasets used for binary classification in L2X (Chen et al., 2018). For each dataset, we have 10k training and 10k test set. In first three datasets, we generate data $\boldsymbol{X}$ from 10-dimensional standard Gaussian and assign labels using $P(Y = 1|\boldsymbol{X}) = 1/(1 + g(\boldsymbol{X}))$ in each dataset, where $g(\boldsymbol{X})$ is defined in the following way: **i) *XOR*:** $exp(f_1 * f_2)$, **ii) *Orange Skin*:** $exp(\sum_{i=1}^{4} f_i{}^2 - 4)$, and **iii) *Nonlinear Additive*:** $exp(-100 * sin(2 * f_1) + 2 * |f_2| + f_3 + exp(-f_4))$. In the fourth dataset, **iv) *Switch*:** We generate $f_{10}$ from a mixture of two Gaussians centered at $\pm 3$ respectively with equal probability. If $f_{10}$ is from the $\mathcal{N}(3, 1)$, then we use $\{f_1, f_2, f_3, f_4\}$ to generate Y from the Orange Skin model. Otherwise, we use $\{f_5, f_6, f_7, f_8\}$ to generate Y from the Nonlinear Additive model. $f_9$ is not used when generating labels.

## B.4  DATA LICENSE

**Aduld Income** and **BlogFeedback** are under Open Data Commons Public Domain Dedication and License (PDDL).

## C   Details of the experiments in the main paper

### C.1   Model architectures and hyper-parameters for XTab

The classifier has three linear layers, two of which are followed by a leakyReLU and dropout (p=0.2). For the mask generator, we use two architectures; i) **shallow:** A linear layer followed by leakyReLU and a final linear layer, ii) **deep:** five linear layers, each followed by leakyReLU, and a final linear layer. The last layer for both mask generator and classifier uses sigmoid activation. The number of hidden units in each layer in the mask generator is same as the number of features in the input while we use 1024 units in each hidden layers of classifier. During training, a learning rate of 0.001 is used for all experiments and we optimize the batch size and total number of epochs.

Table A1: Architectures & hyper-parameters used in our framework for each dataset. Abbreviations are; $\sigma$: Standard deviation used for Gaussian noise, **MR:** Mask ratio $p$.

| Dataset | Mask | Encoder | Classifier | Subsets / Overlap | $\sigma$/MR ($p$) | Noise | Batch/Epoch |
|---|---|---|---|---|---|---|---|
| **SynRank** | [10, 10] | [1024] | [1024, 1024, 1024] | 3 / 75% | 0.5/0.5 | Gaussian | 1024, 40 |
| **SynRank100** | [100, 100] | [1024] | [1024, 1024, 1024] | 2 / 75% | NA/0.5 | Swap | 1024, 40 |
| **Income** | [105, 105] | [1024] | [1024, 1024, 1024] | 3 / 25% | 0.3/0.2 | Gaussian | 1024, 40 |
| **Blog** | [280, 280] | [1024] | [1024, 1024, 1024] | 7 / 75% | 0.3/0.2 | Gaussian | 256, 20 |
| **L2X XOR** | [10, 10] | [1024] | [1024, 1024, 1024] | 2 / 75% | 0.05/0.2 | Gaussian | 1024, 40 |
| **L2X Orange** | [10, 10] | [1024] | [1024, 1024, 1024] | 2 / 75% | 0.05/0.2 | Gaussian | 1024, 40 |
| **L2X N. Additive** | [10, 10] | [1024] | [1024, 1024, 1024] | 2 / 75% | 0.01/0.2 | Gaussian | 1024, 40 |
| **L2X Switch** | [10, 10] | [1024] | [1024, 1024, 1024] | 2 / 75% | 0.05/0.3 | Gaussian | 1024, 40 |

### C.2   Implementation and resources

We implemented our work using PyTorch (Paszke et al., 2019). AdamW optimizer (Loshchilov & Hutter, 2017) with $betas = (0.9, 0.999)$ and $eps = 1e - 07$ is used for all of our experiments. We used a compute cluster consisting of Volta GPUs throughout this work.

### C.3   Details for training L2X, Invase, TabNet, Saliency Maps and Integrated Gradients (IG)

**L2X:** We used the official implementation of L2X[2]. We set all of hyperparameters, following the instruction in L2X paper (Chen et al., 2018). For each data set, we trained a neural network model with three hidden layers. The explainer is a neural network composed of two hidden layers. The variational family is based on three hidden layers. All layers are linear with dimension 200. The number of desired features is set to the number of true features. We fixed the step size to be 0.001 across experiments. The temperature for Gumbel-softmax approximation is fixed to be 0.1. Since the model is proposed for feature selection, we used the average number of selected features for each sample in the test set to rank them.

**Invase:** We followed the hyperparameter selection as instructed in Invase paper (Yoon et al., 2018) and all the experiments are based on the official Keras implementation[3]. We fixed the learning rate and $\lambda$ as 0.0001 and 0.1, respectively. The actor and critic models are three layer neural networks with hidden state dimensions 100 and 200, respectively. L2 regularization is set to be 0.001 and activation function is ReLU. We used the feature selection probability, which is the output of the actor model, to rank the features.

**TabNet:** We used the well established PyTorch implementation of TabNet[4]. We set the hyperparameters as $N_a = N_b = 8$, $\lambda_{sparse} = 0.001$, $B = 1024$, $B_v = 128$, $\gamma = 0.3$, and learning rate $= 0.02$. For all experiments, we used sparsemax as the masking function and OneCycleLR as the learning rate scheduler. The other parameters are set to be same as the default choices.

**Saliency Maps and Integrated Gradients (IG):** For Saliency Maps (Simonyan et al., 2013) and IG (Sundararajan et al., 2017), we used the same architecture as XTab and trained the models using SGD

---

[2]https://github.com/Jianbo-Lab/L2X
[3]https://github.com/jsyoon0823/INVASE
[4]https://github.com/dreamquark-ai/tabnet

with learning rate of 0.01. For Saliency Map, we considered absolute value of each sample gradient for ranking. For IG, we used Captum PyTorch library[5].

# D  MORE ON RELATED WORKS

**Explainability** The literature in explainability and model interpretation is extensive and we refer the reader to the survey papers (Linardatos et al., 2020; Nielsen et al., 2021; Tjoa & Guan, 2020) for a more complete review. In this work, we compare our method to the commonly used methods (Random Forest (Breiman, 2001), Gradient Boosting Classifier (Friedman, 2001; Pedregosa et al., 2011)), to those based on the gradients and/or activations (Saliency Maps (Simonyan et al., 2013) and Integrated Gradients (IG) (Sundararajan et al., 2017)), to the ones that rely on the learnable masks (TabNet (Arık & Pfister, 2021), Invase (Yoon et al., 2018)) and to some of the recently published feature selection methods (L2X (Chen et al., 2018), Invase (Yoon et al., 2018)). What distinguishes our work from the aforementioned works is that we focus on the sensitivity of the feature rankings to model initialisation in neural networks. Our goal is to achieve the robustness of tree-based methods such as Random Forest (Breiman, 2001) in neural network setting. In this regard, we compare our results to neural network-based methods both in the main paper as well as in the Appendix.

**Explainability in Federated Learning** There is some recent work in the intersection of explainable AI (XAI) and Federated Learning (FL) such as the application of Gradient-weighted Class Activation Mapping (Grad-CAM) (Selvaraju et al., 2017) to explain the classification results in electrocardiography (ECG) monitoring healthcare system (Raza et al., 2022). However, it still remains to be an open problem. Lastly, although our method is not proposed for Federated Learning setting, we believe that it can still be used in this area.

# E  ABLATION STUDY ON ARCHITECTURE

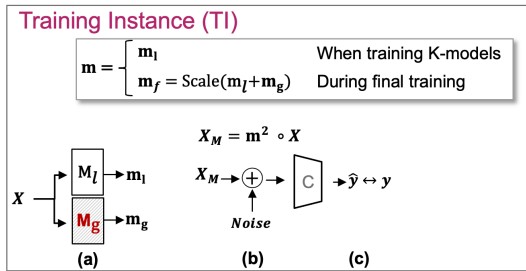

Figure A1: Standard architecture that uses only mask generator and a simple MLP-based classifier.

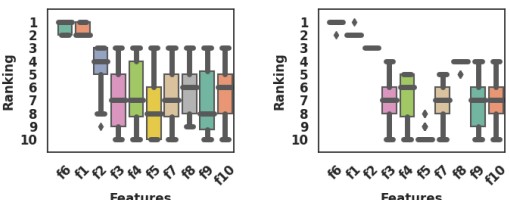

Figure A2: Repeating the experiment in Figure 4a, using a standard architecture shown in Figure A1. The important features are $f_6 > f_1 > f_2$ for SynRank while the rest of the features is not informative. **Left:** Variations in feature rankings across 20 local mask models **Right:** Variations in feature rankings in 100 global models, each of which is obtained by averaging the parameters of 10 models bootstrapped from 20 local models for the same three cases. Comparing it to our original architecture based on SubTab (Ucar et al., 2021), we see that the robustness of feature ranking achieved via parameter-averaging is agnostic to the choice of architecture used for the downstream task.

---

[5]https://github.com/pytorch/captum

# F ADDITIONAL RESULTS FOR SYNRANK DATASET

## F.1 RANKING COMPARISON OF DIFFERENT MODELS IN TABLE FORMAT

Table A2: Comparison of feature ranking obtained from 10 runs of different methods for SynRank. Expected ranking is **f6**>**f1**>**f2**, and incorrect rankings are shown in bold. This is the same result shown as a figure in Figure 3a in the main paper.

| Runs | 1 | 2 | 3 | 4 | 5 | 6 | 7 | 8 | 9 | 10 | Test Acc. |
|---|---|---|---|---|---|---|---|---|---|---|---|
| | | | | | **SynRank dataset** | | | | | | |
| | **Ranking from XTab (no weight regularization is used)** | | | | | | | | | | |
| 1 | f6 | f1 | f2 | f9 | f5 | f4 | f3 | f8 | f7 | f10 | 0.9971 |
| 2 | f6 | f1 | f2 | f9 | f5 | f3 | f8 | f4 | f7 | f10 | 0.9965 |
| 3 | f6 | f1 | f2 | f5 | f9 | f3 | f4 | f8 | f7 | f10 | 0.9969 |
| 4 | f6 | f1 | f2 | f9 | f5 | f4 | f3 | f8 | f7 | f10 | 0.9951 |
| 5 | f6 | f1 | f2 | f9 | f5 | f4 | f3 | f8 | f10 | f7 | 0.9955 |
| 6 | f6 | f1 | f2 | f5 | f9 | f3 | f4 | f8 | f7 | f10 | 0.9972 |
| 7 | f6 | f1 | f2 | f9 | f5 | f3 | f4 | f8 | f10 | f7 | 0.9964 |
| 8 | f6 | f1 | f2 | f5 | f9 | f4 | f3 | f8 | f7 | f10 | 0.9964 |
| 9 | f6 | f1 | f2 | f5 | f9 | f4 | f3 | f8 | f7 | f10 | 0.9971 |
| 10 | f6 | f1 | f2 | f9 | f5 | f3 | f4 | f8 | f7 | f10 | 0.9973 |
| | **Rankings from GBCP & RFP** | | | | | | | | | | |
| GBCP | f6 | f1 | f2 | f10 | f9 | f8 | f7 | f5 | f4 | f3 | 0.9996 |
| RFP | f6 | f1 | f2 | f10 | f9 | f8 | f7 | f5 | f4 | f3 | 0.9999 |
| | **Rankings from TabNet** | | | | | | | | | | |
| 1 | **f1** | **f6** | **f2** | f5 | f8 | f7 | f9 | f10 | f4 | f3 | 0.9990 |
| 2 | **f1** | **f6** | **f2** | f8 | f7 | f3 | f10 | f5 | f4 | f9 | 0.9988 |
| 3 | f6 | f1 | f2 | f10 | f9 | f3 | f5 | f8 | f4 | f7 | 0.9980 |
| 4 | f6 | f1 | f2 | f4 | f7 | f10 | f9 | f5 | f3 | f8 | 0.9978 |
| 5 | **f6** | **f1** | **f8** | f2 | f3 | f10 | f9 | f4 | f5 | f7 | 0.9984 |
| 6 | **f6** | **f1** | **f3** | f2 | f4 | f8 | f10 | f5 | f9 | f7 | 0.9988 |
| 7 | **f6** | **f1** | **f3** | f2 | f7 | f9 | f10 | f8 | f5 | f4 | 0.9999 |
| 8 | **f1** | **f2** | **f6** | f8 | f9 | f7 | f10 | f3 | f4 | f5 | 0.9963 |
| 9 | f6 | f1 | f2 | f9 | f5 | f8 | f10 | f4 | f7 | f3 | 0.9995 |
| 10 | f6 | f1 | f2 | f7 | f3 | f5 | f10 | f4 | f8 | f9 | 0.9979 |
| | **Rankings from Invase** | | | | | | | | | | |
| 1 | **f1** | **f6** | **f2** | f5 | f8 | f3 | f10 | f9 | f4 | f7 | 0.9988 |
| 2 | **f1** | **f6** | **f2** | f5 | f9 | f4 | f10 | f3 | f7 | f8 | 0.9989 |
| 3 | **f1** | **f6** | **f2** | f3 | f8 | f4 | f5 | f9 | f7 | f10 | 0.9989 |
| 4 | f6 | f1 | f2 | f7 | f8 | f4 | f10 | f9 | f5 | f3 | 0.9986 |
| 5 | f6 | f1 | f2 | f4 | f5 | f8 | f10 | f7 | f9 | f3 | 0.9989 |
| 6 | **f1** | **f6** | **f2** | f3 | f7 | f4 | f8 | f9 | f10 | f5 | 0.9992 |
| 7 | **f1** | **f6** | **f2** | f9 | f7 | f4 | f5 | f10 | f8 | f3 | 0.9985 |
| 8 | **f1** | **f6** | **f2** | f4 | f5 | f9 | f3 | f7 | f10 | f8 | 0.9993 |
| 9 | **f1** | **f6** | **f2** | f10 | f3 | f5 | f7 | f8 | f4 | f9 | 0.9996 |
| 10 | f6 | f1 | f2 | f4 | f3 | f5 | f8 | f9 | f10 | f7 | 0.9991 |
| | **Rankings from L2X** | | | | | | | | | | |
| 1 | **f1** | **f2** | **f6** | f5 | f3 | f4 | f7 | f8 | f9 | f10 | 0.9977 |
| 2 | **f1** | **f2** | **f6** | f3 | f5 | f10 | f4 | f7 | f8 | f9 | 0.9983 |
| 3 | **f1** | **f2** | **f6** | f4 | f10 | f3 | f9 | f5 | f7 | f8 | 0.9969 |
| 4 | **f1** | **f2** | **f6** | f3 | f5 | f10 | f4 | f7 | f8 | f9 | 0.9975 |
| 5 | **f1** | **f2** | **f6** | f3 | f5 | f7 | f4 | f8 | f9 | f10 | 0.9976 |
| 6 | **f1** | **f6** | **f2** | f3 | f4 | f5 | f7 | f8 | f9 | f10 | 0.9978 |
| 7 | **f1** | **f2** | **f6** | f5 | f3 | f4 | f7 | f8 | f9 | f10 | 0.9985 |
| 8 | **f1** | **f2** | **f6** | f3 | f5 | f8 | f4 | f7 | f9 | f10 | 0.9976 |
| 9 | **f1** | **f6** | **f2** | f4 | f10 | f8 | f5 | f3 | f7 | f9 | 0.9984 |
| 10 | **f1** | **f2** | **f6** | f3 | f5 | f8 | f4 | f7 | f9 | f10 | 0.9979 |
| | **Rankings from Saliency Maps** | | | | | | | | | | |
| 1 | f6 | f1 | f2 | f5 | f4 | f7 | f3 | f9 | f10 | f8 | 0.9817 |
| 2 | **f1** | **f6** | **f2** | f10 | f3 | f5 | f7 | f8 | f4 | f9 | 0.9820 |
| 3 | **f1** | **f6** | **f2** | f10 | f3 | f8 | f7 | f9 | f5 | f4 | 0.9816 |
| 4 | f6 | f1 | f2 | f10 | f8 | f4 | f5 | f3 | f7 | f9 | 0.9824 |
| 5 | **f1** | **f6** | **f2** | f9 | f8 | f3 | f10 | f5 | f7 | f4 | 0.9812 |
| 6 | f6 | f1 | f2 | f3 | f5 | f8 | f7 | f4 | f9 | f10 | 0.9817 |
| 7 | **f1** | **f6** | **f2** | f5 | f7 | f4 | f9 | f3 | f10 | f8 | 0.9814 |
| 8 | **f1** | **f6** | **f2** | f5 | f10 | f3 | f4 | f8 | f7 | f9 | 0.9817 |
| 9 | f6 | f1 | f2 | f8 | f5 | f10 | f7 | f9 | f3 | f4 | 0.9821 |
| 10 | f6 | f1 | f2 | f3 | f8 | f7 | f5 | f10 | f4 | f9 | 0.9813 |

## F.2 RESULTS FOR DEEP NETWORK

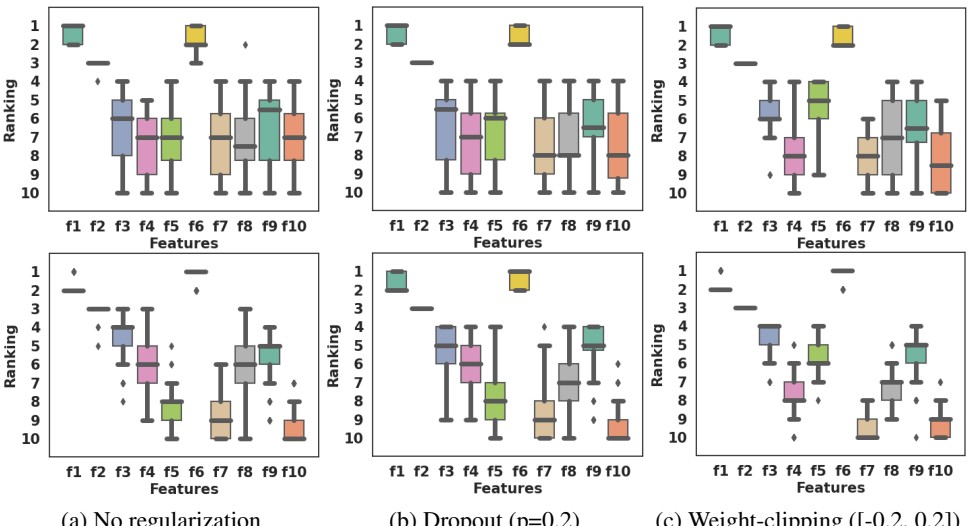

(a) No regularization     (b) Dropout (p=0.2)     (c) Weight-clipping ([-0.2, 0.2])

Figure A3: Measuring the robustness of parameter averaging for deep networks (5 layers). We use SynRank dataset, in which the important features are $f_6 > f_1 > f_2$ while the rest is not informative. **Top row:** Variations in feature rankings across 20 local mask models when we apply **a)** no weight regularization, **b)** Dropout ($p = 0.2$), and **c)** Weight-clipping ($[-0.2, 0.2]$). **Bottom row**: Variations in feature rankings in 100 global models, each of which is obtained by averaging the parameters of 10 models bootstrapped from 20 local models for the same three cases. Weight-clipping helps global models discover important features in the correct order consistently (see the last column, bottom row)

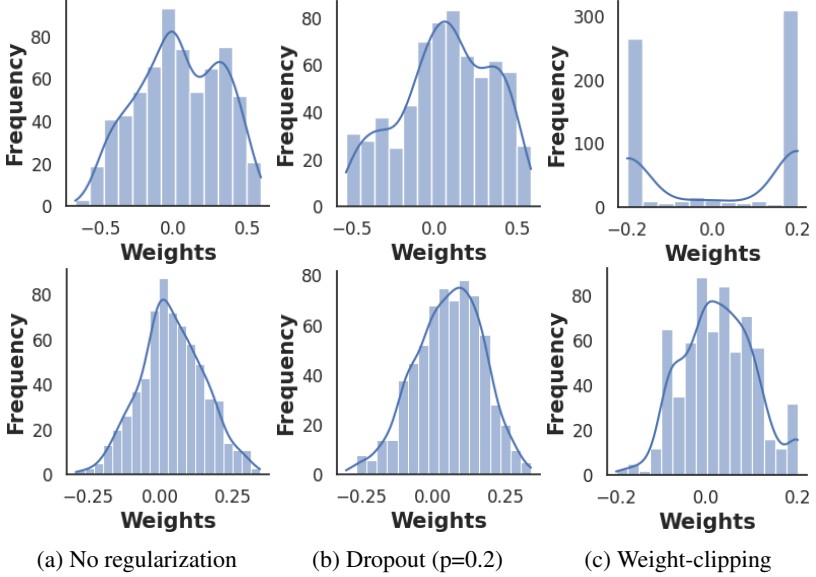

(a) No regularization     (b) Dropout (p=0.2)     (c) Weight-clipping

Figure A4: Measuring the weight distribution for deep networks (5 layers) when we apply **a)** no weight regularization, **b)** Dropout ($p = 0.2$), and **c)** Weight-clipping ($[-0.2, 0.2]$). **Top row:** The weight distribution of a single local mask model selected at random. **Bottom row:** The weight distribution of a global mask model obtained by averaging the parameters of 10 models bootstrapped from 20 local models.

### F.3 RESULTS FOR SHALLOW NETWORK

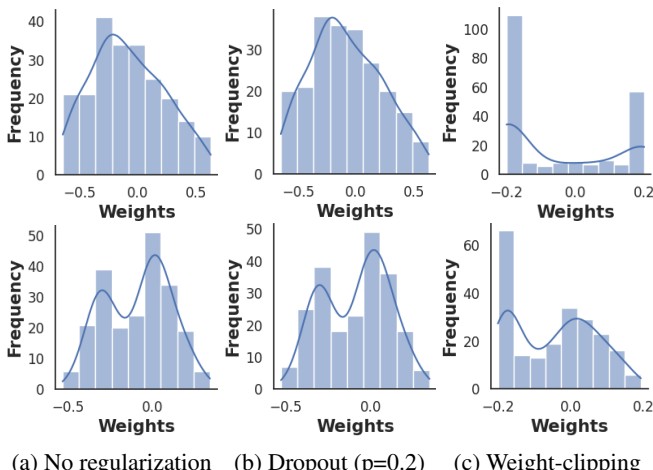

(a) No regularization    (b) Dropout (p=0.2)    (c) Weight-clipping

Figure A5: Measuring the weight distribution for shallow networks when we apply **a)** no weight regularization, **b)** Dropout ($p = 0.2$), and **c)** Weight-clipping ($[-0.2, 0.2]$). **Top row:** The weight distribution of a single local mask model selected at random. **Bottom row:** The weight distribution of a global mask model obtained by averaging the parameters of 10 models bootstrapped from 20 local models.

## G  ADDITIONAL RESULTS FOR SYNRANK100

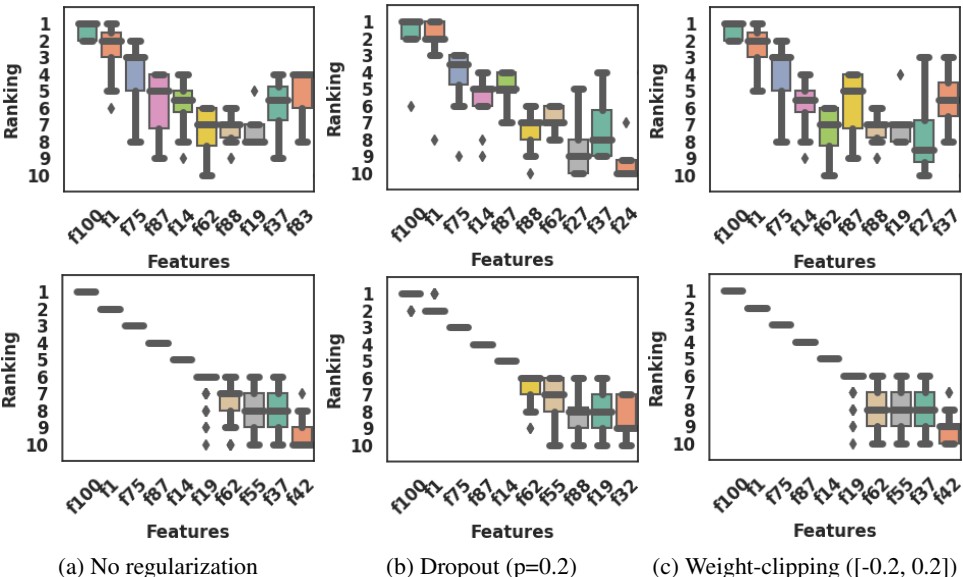

(a) No regularization    (b) Dropout (p=0.2)    (c) Weight-clipping ([-0.2, 0.2])

Figure A6: **SynRank100 dataset:** The important features are $f_{100} > f_1 > f_{75}$ while the rest of the features is not informative. **Top row:** Variations in feature rankings across 20 local mask models when we apply **a)** no weight regularization, **b)** Dropout ($p = 0.2$), and **c)** Weight-clipping ($[-0.2, 0.2]$). **Bottom row**: Variations in feature rankings in 100 global models, each of which is obtained by averaging the parameters of 10 models bootstrapped from 20 local models for the same three cases. We use a shallow network for the mask model.

# H   FEATURE IMPORTANCE RESULTS FOR L2X DATASETS

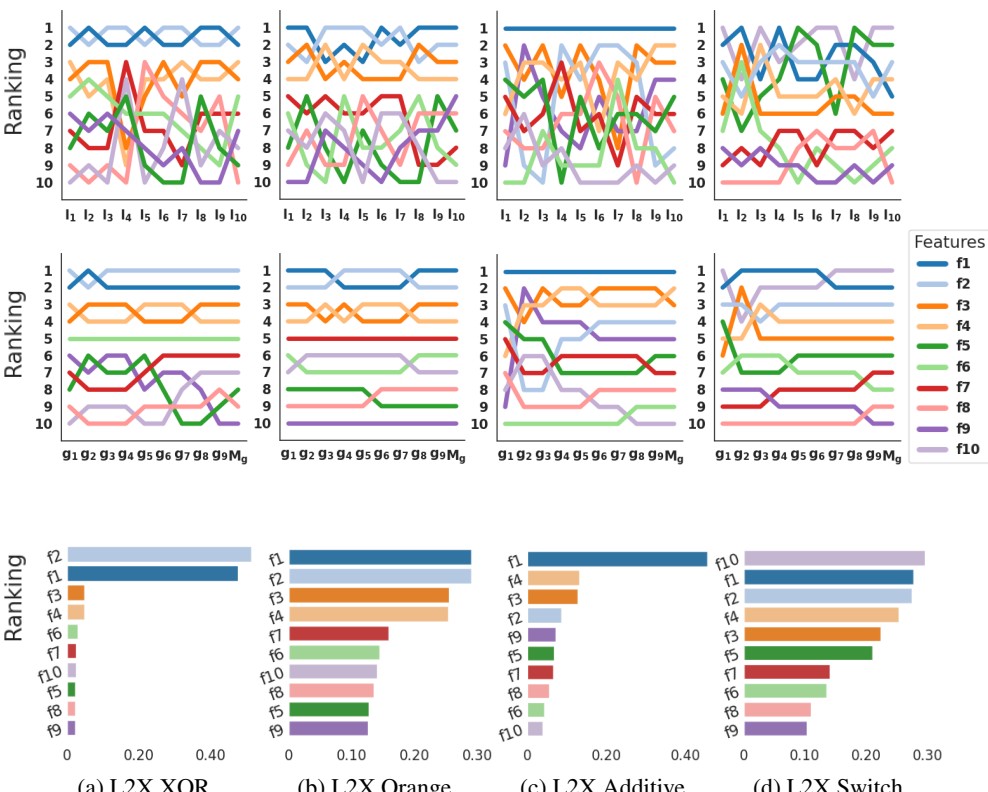

Figure A7: **Top row:** Feature rankings from each of 10 local masks $M_{l_k}$, referred as $l_k$ in the figure, obtained at a particular training run for 10 separate runs. **Middle row**: Feature rankings from the global model, obtained by averaging the parameters of individual models up to a specific run i.e. cumulative average (CA). For example, $g_3$ corresponds to the global model obtained by averaging the parameters of first 3 local masks ($l_1, l_2, l_3$). **Bottom row**: The feature importance weights from $M_g$ obtained by averaging the parameters of all local masks.

We note that the weights of the features are correlated with the frequency and position of their ranks across all local masks. Specifically, $f_1$ and $f_2$ in L2X XOR dataset keep switching positions between $1^{st}$ and $2^{nd}$ ranks across all ten runs (top row in Figure A7(a)). Therefore, $M_g$ computes their importance weights to be similar, giving a slight edge to $f_2$ since it is ranked as #1 by six out of ten local masks (the first and third rows in Figure A7(a)).

In the L2X Switch dataset, $f_{10}$ is used as the switch feature to change whether the label is determined by the features $f_1 - f_4$ or by $f_5 - f_8$ and is discovered as the most important global feature by $M_g$ (the bottom row in Figure A7(d)). Please note that this is the hardest synthetic dataset in a way that the effect of $f_{10}$ on the sample labels is not direct, rather it influences the labels indirectly by deciding which features to be used for label generation. This might be the main reason why a commonly used method such as permutation feature importance fails, ranking $f_1$ as the most important feature in our experiments with random forest and gradient boosting classifier used together with permutation feature importance (please see the results in Section J.1 of the Appendix). Consistent with the ground truth, $M_g$ also discovers $f_9$ as an uninformative feature (bottom row in Figure A7(d)).

In L2X Orange dataset, our method correctly discovers the first four features as the most important ones with almost equal weights while it indicates $f_1$ and $f_4$ as the most important features in L2X Nonlinear Additive.

# I    ADDITIONAL RESULTS FOR L2X SWITCH

## I.1    RESULTS FOR SHALLOW NETWORK

We run additional experiments on L2X Switch under three conditions: We apply i) No weight regularization to the weights of the mask model, ii) Dropout with $p = 0.2$ for the layers with leaky ReLU activation, iii) Weight-clipping ($[-0.2, 0.2]$) to limit the magnitude of the weights in each layer. For each of the three cases, we train 20 separate models and compare two different settings. In the first setting, we compute the variation in the feature rankings given by 20 local mask models (top row in Figure A8). In the second setting, we obtain 100 global models, each of which is obtained by averaging the parameters of 10 local models *bootstrapped* from 20 models. We compare the variation in feature rankings given by global models (second setting – bottom row in Figure A8) to that of 20 local models (first setting – top row). We observe that: i) Regularization methods such as dropout and weight-clipping help improve the variation across 20 local models, but the improvement is not substantial (e.g., comparing $f_9$ and $f_{10}$ across three cases at the top row). ii) However, they help improve the robustness of the parameter averaging significantly (e.g., comparing $f_9$ and $f_{10}$ across three cases at the bottom row). We also observe that the parameter averaging itself pushes the magnitude of the weights towards a tight range around zero as shown in Figure A9 (Section I.1 of the Appendix), indicating a potential relationship between robustness and a tighter weight distribution. Overall, the global models are able to discover important features in the correct order, and the variation in feature ranking across global models is small (almost zero) for ground truth important features when we apply weight regularization (please see $f_9$ and $f_{10}$ at the bottom row in Figure A8b and Figure A8c). iii) Weight-clipping works better than dropout in our experiments, but dropout results could perhaps be improved by hyper-parameter search on the $p$ variable.

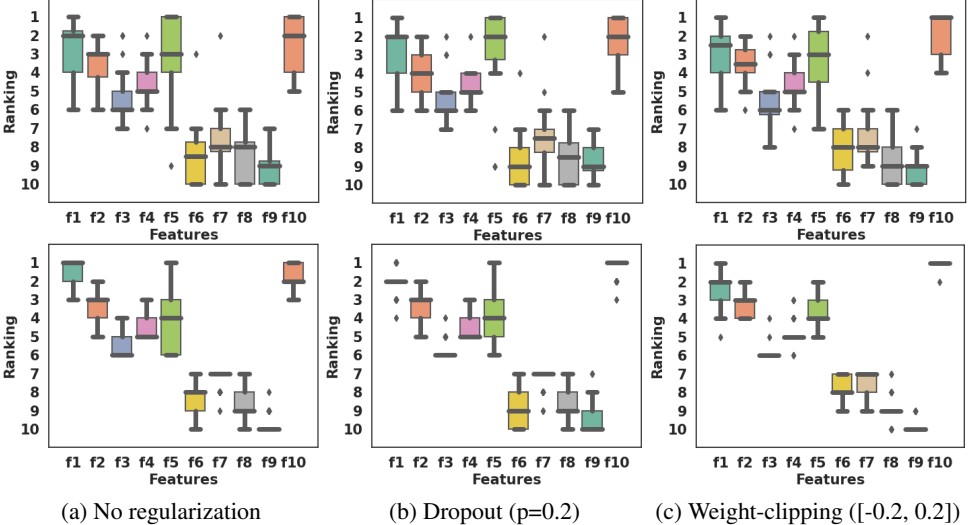

(a) No regularization    (b) Dropout (p=0.2)    (c) Weight-clipping ([-0.2, 0.2])

Figure A8: Measuring the robustness of parameter averaging for shallow networks. We use L2X Switch dataset, in which the most and the least important features are $f_{10}$ and $f_9$ respectively. **Top row:** Variations in feature rankings across 20 local mask models when we apply **a)** no weight regularization, **b)** Dropout ($p = 0.2$), and **c)** Weight-clipping ($[-0.2, 0.2]$). **Bottom row**: Variations in feature rankings in 100 global models, each of which is obtained by averaging the parameters of 10 models bootstrapped from 20 local models for the same three cases.

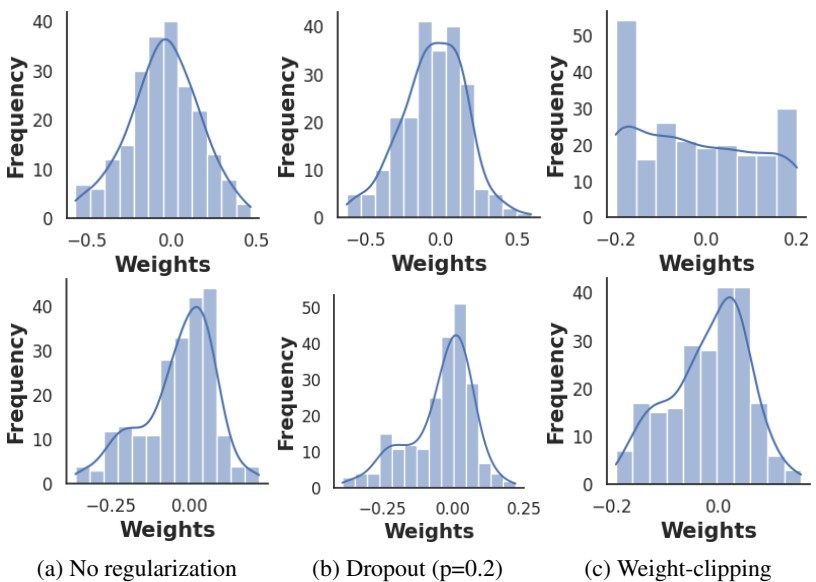

(a) No regularization     (b) Dropout (p=0.2)     (c) Weight-clipping

Figure A9: The weight distribution for shallow networks when we apply **a)** no weight regularization, **b)** Dropout ($p = 0.2$), and **c)** Weight-clipping ($[-0.2, 0.2]$). **Top row:** The weight distribution of a single local mask model selected at random. **Bottom row:** The weight distribution of a global mask model obtained by averaging the parameters of 10 models bootstrapped from 20 local models.

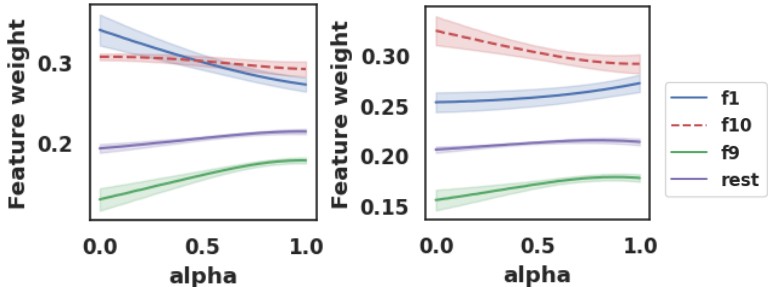

Figure A10: Two separate examples showing the weights of the features from a model that is obtained by interpolating between a single local model and the global model, which is obtained by bootstrapping 10 local models out of 20: $\theta^* = (1-\alpha) * \theta_l + \alpha * \theta_g$, where $\alpha$ is swept from 0 to 1 in 50 steps. The figure on the left uses a local model that ranks features as $f1 > f10 > rest > f9$ while the one on the right uses a local model that ranks features as $f10 > f1 > rest > f9$. As we move from local model ($\alpha = 0$) to the global model ($\alpha = 1$), the estimate of feature ranking gets better (Expected feature rank: $f10 > f1 > rest > f9$). The plots show the mean and 95% confidence intervals for the weights of the features obtained by repeating the experiments with 100 bootstrapped global models, each of which is interpolated with the same local model. In both examples, we see that the global model predicts the rankings of the features correctly even when the initial ranking from the local model might be incorrect (e.g., left plot). **"rest"** refers to average weight of the rest of the informative features i.e., $f2 - f8$.

### I.2 RESULTS FOR DEEPER NETWORK

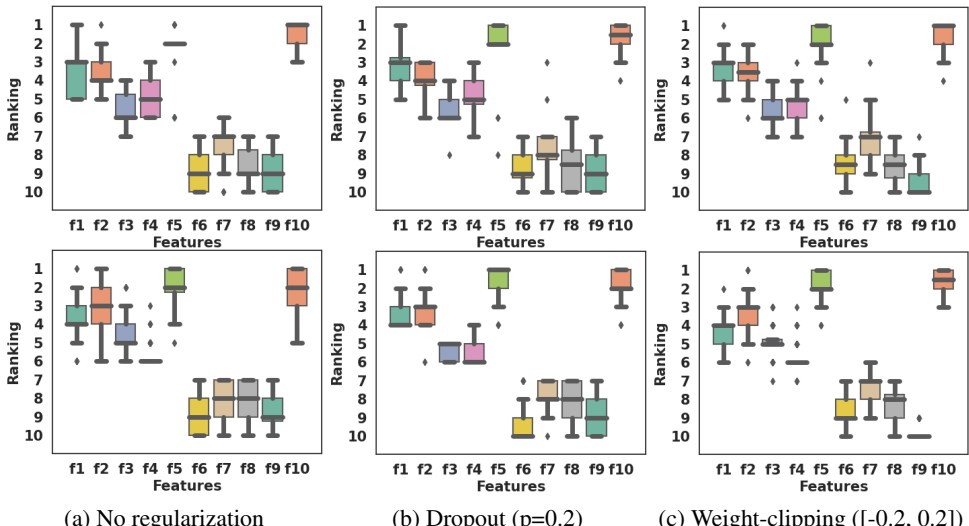

(a) No regularization  (b) Dropout (p=0.2)  (c) Weight-clipping ([-0.2, 0.2])

Figure A11: Measuring the robustness of parameter averaging for deeper networks (5 layers). We use L2X Switch dataset, in which the most and the least important features are $f_{10}$ and $f_9$ respectively. **Top row:** Variations in feature rankings across 20 local mask models when we apply **a)** no weight regularization, **b)** Dropout ($p = 0.2$), and **c)** Weight-clipping ($[-0.2, 0.2]$). **Bottom row:** Variations in feature rankings in 100 global models, each of which is obtained by averaging the parameters of 10 models bootstrapped from 20 local models for the same three cases. Although the weight regularization helps improve the robustness, we still see variations for feature $f_{10}$ when the model is deep (see last two columns at the bottom row). Also, weight-clipping works better than dropout again.

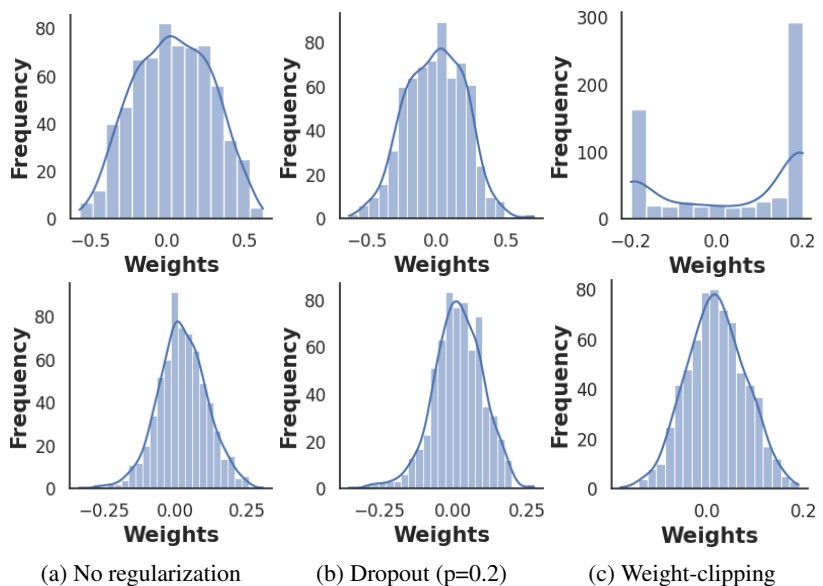

(a) No regularization  (b) Dropout (p=0.2)  (c) Weight-clipping

Figure A12: Measuring the weight distribution for deep networks (5 layers) when we apply **a)** no weight regularization, **b)** Dropout ($p = 0.2$), and **c)** Weight-clipping ($[-0.2, 0.2]$). **Top row:** The weight distribution of a single local mask model selected at random. **Bottom row:** The weight distribution of a global mask model obtained by averaging the parameters of 10 models bootstrapped from 20 local models.

## J    MORE RESULTS FOR SYNTHETIC DATASETS FROM L2X

In this section, we show the robustness of our method, **XTab**, by listing the global feature importance from $M_g$ using *test set* for L2X (Chen et al., 2018) datasets across 10 separate runs of our method. *Please note that we don't use weight regularization for XTab in any of these experiments.* We also list global feature importance obtained by using permutation importance together with random forest (RFP), and gradient boosting classifier (GBCP). For L2X Nonlinear Additive and L2X Switch datasets, we also compare **XTab** to other approaches such as TabNet(Arık & Pfister, 2021), Invase(Yoon et al., 2018), L2X(Chen et al., 2018), Saliency Maps(Simonyan et al., 2013), and Integrated Gradients(Sundararajan et al., 2017) for comparison.

### J.1    COMPARING XTAB TO OTHER METHODS USING L2X SWITCH DATASET

Table A3: Comparing XTab to other methods using L2X Switch dataset by listing global feature importance rankings across 10 separate runs with different set of random seeds.

| | | | | | | **L2X Switch** | | | | | |
|---|---|---|---|---|---|---|---|---|---|---|---|
| | | | | **Ranking from XTab (no weight regularization is used)** | | | | | | | |
| **Runs** | 1 | 2 | 3 | 4 | 5 | 6 | 7 | 8 | 9 | 10 | **Test Acc.** |
| 1 | f10 | f1 | f2 | f4 | f5 | f3 | f7 | f6 | f8 | f9 | 0.9768 |
| 2 | f10 | f2 | f5 | f1 | f4 | f3 | f7 | f8 | f6 | f9 | 0.975 |
| 3 | f10 | f1 | f2 | f5 | f4 | f3 | f7 | f8 | f6 | f9 | 0.9742 |
| 4 | f10 | f1 | f2 | f4 | f5 | f3 | f7 | f6 | f8 | f9 | 0.9733 |
| 5 | f10 | f5 | f1 | f2 | f4 | f3 | f6 | f7 | f8 | f9 | 0.9737 |
| 6 | f10 | f1 | f4 | f5 | f2 | f3 | f7 | f6 | f8 | f9 | 0.9741 |
| 7 | f10 | f1 | f2 | f5 | f4 | f3 | f7 | f9 | f8 | f6 | 0.9729 |
| 8 | f10 | f1 | f2 | f4 | f5 | f3 | f7 | f8 | f6 | f9 | 0.9682 |
| 9 | f10 | f1 | f5 | f4 | f2 | f3 | f7 | f6 | f8 | f9 | 0.9725 |
| 10 | f10 | f1 | f2 | f4 | f5 | f3 | f7 | f6 | f8 | f9 | 0.9757 |
| **GBCP** | f1 | f10 | f4 | f2 | f3 | f5 | f7 | f8 | f6 | f9 | 0.9676 |
| **RFP** | f1 | f10 | f5 | f4 | f3 | f2 | f8 | f7 | f6 | f9 | 0.9575 |
| | | | | | **Ranking from TabNet** | | | | | | |
| **Runs** | 1 | 2 | 3 | 4 | 5 | 6 | 7 | 8 | 9 | 10 | **Test Acc.** |
| 1 | f4 | f1 | f10 | f2 | f3 | f5 | f6 | f8 | f9 | f7 | 0.9606 |
| 2 | f5 | f10 | f4 | f1 | f2 | f3 | f7 | f9 | f8 | f6 | 0.9733 |
| 3 | f4 | f1 | f10 | f2 | f3 | f9 | f8 | f5 | f7 | f6 | 0.9646 |
| 4 | f1 | f4 | f2 | f10 | f5 | f3 | f6 | f7 | f9 | f8 | 0.9723 |
| 5 | f10 | f5 | f2 | f4 | f3 | f7 | f1 | f9 | f6 | f8 | 0.9623 |
| 6 | f1 | f2 | f4 | f10 | f3 | f5 | f9 | f6 | f8 | f7 | 0.9669 |
| 7 | f1 | f2 | f4 | f3 | f10 | f5 | 6 | f8 | f7 | f9 | 0.9723 |
| 8 | f10 | f4 | f3 | f1 | f5 | f2 | f8 | f7 | f6 | f9 | 0.9683 |
| 9 | f4 | f1 | f2 | f10 | f3 | f5 | f9 | f6 | f7 | f8 | 0.9695 |
| 10 | f1 | f10 | f3 | f4 | f2 | f5 | f8 | f7 | f6 | f9 | 0.9723 |
| | | | | | **Ranking from Invase** | | | | | | |
| **Runs** | 1 | 2 | 3 | 4 | 5 | 6 | 7 | 8 | 9 | 10 | **Test Acc.** |
| 1 | f4 | f10 | f2 | f1 | f3 | f5 | f6 | f8 | f9 | f7 | 0.982 |
| 2 | f3 | f4 | f10 | f2 | f1 | f5 | f9 | f8 | f6 | f7 | 0.978 |
| 3 | f1 | f10 | f3 | f2 | f4 | f5 | f8 | f9 | f7 | f6 | 0.979 |
| 4 | f10 | f1 | f2 | f3 | f4 | f5 | f8 | f9 | f6 | f7 | 0.979 |
| 5 | f10 | f2 | f1 | f3 | f4 | f5 | f9 | f6 | f8 | f7 | 0.981 |
| 6 | f10 | f2 | f4 | f1 | f3 | f5 | f9 | f8 | f6 | f7 | 0.979 |
| 7 | f10 | f4 | f3 | f1 | f2 | f5 | f7 | f6 | f9 | f8 | 0.982 |
| 8 | f4 | f3 | f10 | f1 | f2 | f5 | f9 | f7 | f8 | f6 | 0.981 |
| 9 | f10 | f1 | f4 | f3 | f2 | f5 | f6 | f7 | f9 | f8 | 0.982 |
| 10 | f10 | f2 | f4 | f1 | f3 | f5 | f7 | f8 | f6 | f9 | 0.982 |
| | | | | | **Ranking from L2X** | | | | | | |
| **Runs** | 1 | 2 | 3 | 4 | 5 | 6 | 7 | 8 | 9 | 10 | **Test Acc.** |
| 1 | f1 | f4 | f7 | f3 | f5 | f2 | f10 | f9 | f6 | f8 | 0.9869 |
| 2 | f1 | f9 | f2 | f7 | f3 | f4 | f10 | f8 | f6 | f5 | 0.9913 |
| 3 | f1 | f4 | f2 | f3 | f9 | f5 | f10 | f6 | f7 | f8 | 0.9943 |
| 4 | f1 | f4 | f2 | f8 | f3 | f10 | f9 | f7 | f6 | f5 | 0.9876 |
| 5 | f1 | f3 | f2 | f4 | f10 | f8 | f7 | f5 | f6 | f9 | 0.9891 |
| 6 | f1 | f10 | f5 | f2 | f4 | f3 | f8 | f7 | f6 | f9 | 0.9925 |
| 7 | f1 | f4 | f3 | f5 | f2 | f7 | f6 | f10 | f9 | f8 | 0.9896 |
| 8 | f1 | f4 | f8 | f2 | f3 | f10 | f9 | f6 | f7 | f5 | 0.9905 |
| 9 | f1 | f4 | f2 | f8 | f3 | f7 | f9 | f6 | f10 | f5 | 0.9923 |
| 10 | f1 | f4 | f2 | f3 | f6 | f10 | f7 | f5 | f8 | f9 | 0.9906 |
| | | | | | **Ranking from Saliency Maps** | | | | | | |
| **Runs** | 1 | 2 | 3 | 4 | 5 | 6 | 7 | 8 | 9 | 10 | **Test Acc.** |
| 1 | f1 | f10 | f5 | f4 | f2 | f3 | f8 | f7 | f6 | f9 | 0.9334 |
| 2 | f1 | f10 | f5 | f2 | f4 | f3 | f7 | f6 | f8 | f9 | 0.9355 |
| 3 | f1 | f10 | f5 | f4 | f2 | f3 | f7 | f6 | f8 | f9 | 0.9378 |
| 4 | f1 | f10 | f5 | f4 | f2 | f3 | f7 | f6 | f8 | f9 | 0.9372 |
| 5 | f1 | f10 | f5 | f4 | f2 | f3 | f7 | f8 | f6 | f9 | 0.9372 |
| 6 | f1 | f10 | f5 | f4 | f2 | f3 | f7 | f8 | f6 | f9 | 0.9378 |
| 7 | f1 | f10 | f5 | f4 | f2 | f3 | f7 | f6 | f8 | f9 | 0.9346 |
| 8 | f1 | f10 | f5 | f4 | f2 | f3 | f7 | f8 | f6 | f9 | 0.9337 |
| 9 | f1 | f10 | f5 | f4 | f2 | f3 | f7 | f8 | f6 | f9 | 0.9377 |
| 10 | f1 | f10 | f5 | f2 | f4 | f3 | f8 | f7 | f6 | f9 | 0.9331 |
| | | | | | **Ranking from Integrated Gradients (IG)** | | | | | | |
| **Runs** | 1 | 2 | 3 | 4 | 5 | 6 | 7 | 8 | 9 | 10 | **Test Acc.** |
| 1 | f2 | f5 | f1 | f4 | f8 | f3 | f6 | f7 | f10 | f9 | 0.9334 |
| 2 | f2 | f5 | f1 | f3 | f4 | f6 | f7 | f8 | f10 | f9 | 0.9355 |
| 3 | f2 | f5 | f1 | f3 | f4 | f8 | f6 | f10 | f7 | f9 | 0.9378 |
| 4 | f2 | f5 | f1 | f3 | f6 | f4 | f7 | f8 | f10 | f9 | 0.9372 |
| 5 | f2 | f5 | f4 | f1 | f3 | f6 | f8 | f10 | f7 | f9 | 0.9372 |
| 6 | f2 | f5 | f4 | f1 | f8 | f3 | f10 | f7 | f6 | f9 | 0.9378 |
| 7 | f2 | f5 | f1 | f3 | f4 | f10 | f7 | f8 | f6 | f9 | 0.9346 |
| 8 | f2 | f5 | f1 | f4 | f3 | f6 | f8 | f7 | f10 | f9 | 0.9337 |
| 9 | f5 | f2 | f4 | f1 | f8 | f3 | f6 | f7 | f10 | f9 | 0.9377 |
| 10 | f2 | f5 | f1 | f4 | f3 | f8 | f6 | f10 | f7 | f9 | 0.9331 |

## J.2 Comparing XTab to other methods using L2X Non-Linear Additive dataset

Table A4: Comparing XTab to other methods using L2X Non-Linear Additive dataset by listing feature importance rankings across 10 separate runs with different set of random seeds.

| | L2X Nonlinear Additive | | | | | | | | | | |
|---|---|---|---|---|---|---|---|---|---|---|---|
| | **Ranking from XTab (no weight regularization is used)** | | | | | | | | | | |
| **Runs** | 1 | 2 | 3 | 4 | 5 | 6 | 7 | 8 | 9 | 10 | **Test Acc.** |
| 1 | f1 | f4 | f3 | f5 | f7 | f9 | f2 | f8 | f6 | f10 | 0.9805 |
| 2 | f1 | f4 | f3 | f5 | f9 | f7 | f8 | f2 | f6 | f10 | 0.9775 |
| 3 | f1 | f4 | f3 | f5 | f8 | f2 | f9 | f7 | f10 | f6 | 0.9814 |
| 4 | f1 | f4 | f3 | f2 | f5 | f9 | f7 | f8 | f6 | f10 | 0.9854 |
| 5 | f1 | f4 | f3 | f5 | f7 | f9 | f8 | f6 | f2 | f10 | 0.9844 |
| 6 | f1 | f4 | f3 | f5 | f9 | f7 | f8 | f2 | f6 | f10 | 0.9835 |
| 7 | f1 | f4 | f3 | f7 | f5 | f8 | f9 | f2 | f6 | f10 | 0.9822 |
| 8 | f1 | f4 | f3 | f5 | f9 | f7 | f2 | f6 | f10 | f8 | 0.9761 |
| 9 | f1 | f4 | f3 | f5 | f7 | f9 | f8 | f2 | f6 | f10 | 0.9835 |
| 10 | f1 | f4 | f3 | f5 | f9 | f2 | f7 | f6 | f8 | f10 | 0.9775 |
| **GBCP** | f1 | f4 | f3 | f2 | f10 | f6 | f5 | f9 | f8 | f7 | 0.9924 |
| **RFP** | f1 | f4 | f3 | f2 | f10 | f5 | f7 | f9 | f6 | f8 | 0.9853 |
| | **Ranking from TabNet** | | | | | | | | | | |
| **Runs** | 1 | 2 | 3 | 4 | 5 | 6 | 7 | 8 | 9 | 10 | **Test Acc.** |
| 1 | f1 | f4 | f5 | f6 | f8 | f10 | f2 | f7 | f3 | f9 | 0.9785 |
| 2 | f1 | f4 | f5 | f6 | f3 | f8 | f9 | f7 | f2 | f10 | 0.9752 |
| 3 | f1 | f5 | f4 | f6 | f8 | f10 | f2 | f3 | f9 | f7 | 0.9751 |
| 4 | f1 | f4 | f5 | f6 | f8 | f3 | f2 | f10 | f7 | f9 | 0.9785 |
| 5 | f1 | f5 | f4 | f6 | f8 | f10 | f2 | f3 | f9 | f7 | 0.9737 |
| 6 | f1 | f5 | f4 | f3 | f6 | f8 | f2 | f10 | f9 | f7 | 0.9758 |
| 7 | f1 | f5 | f4 | f6 | f3 | f2 | f8 | f10 | f7 | f9 | 0.9760 |
| 8 | f1 | f4 | f5 | f6 | f8 | f10 | f2 | f7 | f3 | f9 | 0.9758 |
| 9 | f1 | f4 | f6 | f5 | f2 | f10 | f8 | f3 | f9 | f7 | 0.9758 |
| 10 | f1 | f5 | f4 | f6 | f3 | f2 | f8 | f10 | f7 | f9 | 0.9751 |
| | **Ranking from Invase** | | | | | | | | | | |
| **Runs** | 1 | 2 | 3 | 4 | 5 | 6 | 7 | 8 | 9 | 10 | **Test Acc.** |
| 1 | f1 | f4 | f3 | f2 | f7 | f9 | f8 | f10 | f6 | f5 | 0.987 |
| 2 | f1 | f4 | f3 | f2 | f5 | f9 | f6 | f10 | f7 | f8 | 0.987 |
| 3 | f1 | f4 | f3 | f2 | f8 | f10 | f9 | f6 | f5 | f7 | 0.986 |
| 4 | f1 | f4 | f3 | f2 | f9 | f8 | f7 | f5 | f6 | f10 | 0.987 |
| 5 | f1 | f3 | f4 | f2 | f8 | f7 | f6 | f5 | f9 | f10 | 0.987 |
| 6 | f1 | f4 | f3 | f2 | f9 | f8 | f10 | f5 | f6 | f7 | 0.988 |
| 7 | f1 | f4 | f3 | f2 | f8 | f7 | f10 | f6 | f9 | f5 | 0.988 |
| 8 | f1 | f4 | f3 | f2 | f8 | f9 | f6 | f7 | f5 | f10 | 0.988 |
| 9 | f1 | f4 | f3 | f2 | f9 | f10 | f8 | f5 | f7 | f6 | 0.986 |
| 10 | f1 | f4 | f3 | f2 | f7 | f5 | f8 | f9 | f10 | f6 | 0.986 |
| | **Ranking from L2X** | | | | | | | | | | |
| **Runs** | 1 | 2 | 3 | 4 | 5 | 6 | 7 | 8 | 9 | 10 | **Test Acc.** |
| 1 | f1 | f4 | f5 | f6 | f8 | f10 | f2 | f7 | f3 | f9 | 0.9785 |
| 2 | f1 | f4 | f5 | f6 | f3 | f8 | f9 | f7 | f2 | f10 | 0.9752 |
| 3 | f1 | f5 | f4 | f6 | f8 | f10 | f2 | f3 | f9 | f7 | 0.9751 |
| 4 | f1 | f4 | f5 | f6 | f8 | f3 | f2 | f10 | f7 | f9 | 0.9785 |
| 5 | f1 | f5 | f4 | f6 | f8 | f10 | f2 | f3 | f9 | f7 | 0.9737 |
| 6 | f1 | f5 | f4 | f3 | f6 | f8 | f2 | f10 | f9 | f7 | 0.9758 |
| 7 | f1 | f5 | f4 | f6 | f3 | f2 | f8 | f10 | f7 | f9 | 0.9760 |
| 8 | f1 | f4 | f5 | f6 | f8 | f10 | f2 | f7 | f3 | f9 | 0.9758 |
| 9 | f1 | f4 | f6 | f5 | f2 | f10 | f8 | f3 | f9 | f7 | 0.9758 |
| 10 | f1 | f5 | f4 | f6 | f3 | f2 | f8 | f10 | f7 | f9 | 0.9751 |
| | **Ranking from Saliency Maps** | | | | | | | | | | |
| **Runs** | 1 | 2 | 3 | 4 | 5 | 6 | 7 | 8 | 9 | 10 | **Test Acc.** |
| 1 | f1 | f3 | f2 | f4 | f10 | f5 | f7 | f9 | f8 | f6 | 0.9884 |
| 2 | f1 | f3 | f2 | f4 | f5 | f10 | f9 | f6 | f7 | f8 | 0.9870 |
| 3 | f1 | f3 | f4 | f2 | f10 | f9 | f5 | f6 | f7 | f8 | 0.9876 |
| 4 | f1 | f3 | f4 | f5 | f7 | f10 | f2 | f6 | f8 | f9 | 0.9878 |
| 5 | f1 | f3 | f2 | f4 | f5 | f7 | f10 | f6 | f9 | f8 | 0.9880 |
| 6 | f1 | f3 | f4 | f2 | f10 | f7 | f5 | f6 | f9 | f8 | 0.9874 |
| 7 | f1 | f3 | f2 | f4 | f7 | f9 | f10 | f6 | f5 | f8 | 0.9881 |
| 8 | f1 | f3 | f4 | f5 | f2 | f10 | f6 | f9 | f7 | f8 | 0.9869 |
| 9 | f1 | f3 | f4 | f5 | f2 | f10 | f6 | f9 | f7 | f8 | 0.9883 |
| 10 | f1 | f3 | f4 | f7 | f10 | f5 | f2 | f9 | f6 | f8 | 0.9880 |
| | **Ranking from Integrated Gradients (IG)** | | | | | | | | | | |
| **Runs** | 1 | 2 | 3 | 4 | 5 | 6 | 7 | 8 | 9 | 10 | **Test Acc.** |
| 1 | f1 | f4 | f3 | f2 | f8 | f5 | f9 | f6 | f10 | f7 | 0.9884 |
| 2 | f1 | f4 | f3 | f2 | f5 | f7 | f9 | f6 | f10 | f8 | 0.9870 |
| 3 | f1 | f4 | f3 | f2 | f6 | f9 | f7 | f10 | f8 | f5 | 0.9876 |
| 4 | f1 | f4 | f3 | f8 | f2 | f6 | f10 | f7 | f9 | f5 | 0.9878 |
| 5 | f1 | f4 | f3 | f9 | f2 | f8 | f6 | f10 | f5 | f7 | 0.9880 |
| 6 | f1 | f4 | f3 | f2 | f9 | f8 | f6 | f10 | f7 | f5 | 0.9874 |
| 7 | f1 | f4 | f3 | f10 | f2 | f6 | f9 | f8 | f7 | f5 | 0.9881 |
| 8 | f1 | f4 | f3 | f5 | f2 | f6 | f10 | f9 | f8 | f7 | 0.9869 |
| 9 | f1 | f4 | f3 | f2 | f9 | f8 | f5 | f10 | f6 | f7 | 0.9883 |
| 10 | f1 | f4 | f3 | f2 | f6 | f9 | f10 | f5 | f7 | f8 | 0.9880 |

### J.3 THE RESULTS FROM L2X XOR AND ORANGE DATASETS FOR XTAB

Table A5: Showing the robustness of our method, **XTab**, by listing the global feature importance from $M_g$ using *test set* for L2X XOR and Orange datasets across 10 separate runs of our method. We also listed global feature importance obtained by using permutation importance together with random forest (RFP), and gradient boosting classifier (GBCP) for comparison. In L2X XOR, we expect $f_1$ and $f_2$ to be equally likely to be top 1 and 2 features and vice versa. For L2X Orange, we expect the first four features to be the top four features in no particular order.

**L2X XOR**

| Runs | \multicolumn{10}{c}{Ranking from XTab (no weight regularization is used)} | | | | | | | | | | Test Acc. |
|---|---|---|---|---|---|---|---|---|---|---|---|
| | 1 | 2 | 3 | 4 | 5 | 6 | 7 | 8 | 9 | 10 | |
| 1 | f1 | f2 | f4 | f3 | f6 | f8 | f5 | f7 | f9 | f10 | 0.9911 |
| 2 | f1 | f2 | f4 | f3 | f5 | f6 | f8 | f7 | f9 | f10 | 0.9924 |
| 3 | f1 | f2 | f4 | f3 | f8 | f5 | f6 | f7 | f9 | f10 | 0.9922 |
| 4 | f1 | f2 | f4 | f3 | f5 | f8 | f6 | f9 | f7 | f10 | 0.9865 |
| 5 | f1 | f2 | f4 | f3 | f5 | f8 | f6 | f7 | f9 | f10 | 0.9843 |
| 6 | f1 | f2 | f4 | f3 | f8 | f5 | f9 | f6 | f7 | f10 | 0.9926 |
| 7 | f1 | f2 | f4 | f3 | f8 | f7 | f5 | f6 | f9 | f10 | 0.986 |
| 8 | f1 | f2 | f4 | f3 | f5 | f6 | f8 | f7 | f9 | f10 | 0.9797 |
| 9 | f1 | f2 | f4 | f3 | f5 | f8 | f7 | f6 | f9 | f10 | 0.9915 |
| 10 | f1 | f2 | f4 | f3 | f6 | f5 | f8 | f9 | f7 | f10 | 0.989 |
| GBCP | f1 | f2 | f10 | f9 | f8 | f7 | f6 | f5 | f4 | f3 | 0.9999 |
| RFP | f1 | f2 | f8 | f7 | f9 | f10 | f4 | f3 | f6 | f5 | 0.9995 |

**L2X Orange**

| Runs | \multicolumn{10}{c}{Ranking from XTab (no weight regularization is used)} | | | | | | | | | | Test Acc. |
|---|---|---|---|---|---|---|---|---|---|---|---|
| | 1 | 2 | 3 | 4 | 5 | 6 | 7 | 8 | 9 | 10 | |
| 1 | f1 | f4 | f2 | f3 | f7 | f6 | f8 | f9 | f5 | f10 | 0.954 |
| 2 | f1 | f4 | f2 | f3 | f7 | f9 | f8 | f6 | f5 | f10 | 0.9471 |
| 3 | f1 | f4 | f2 | f3 | f8 | f7 | f9 | f6 | f5 | f10 | 0.9695 |
| 4 | f1 | f4 | f2 | f3 | f9 | f5 | f8 | f6 | f7 | f10 | 0.9747 |
| 5 | f1 | f4 | f2 | f3 | f7 | f8 | f5 | f9 | f6 | f10 | 0.9598 |
| 6 | f1 | f4 | f2 | f3 | f8 | f7 | f9 | f5 | f6 | f10 | 0.9689 |
| 7 | f1 | f4 | f2 | f3 | f8 | f7 | f9 | f6 | f5 | f10 | 0.9658 |
| 8 | f1 | f4 | f2 | f3 | f7 | f6 | f5 | f8 | f9 | f10 | 0.9718 |
| 9 | f1 | f4 | f2 | f3 | f8 | f7 | f9 | f5 | f6 | f10 | 0.9691 |
| 10 | f1 | f4 | f2 | f3 | f8 | f9 | f7 | f6 | f5 | f10 | 0.9735 |
| GBCP | f3 | f1 | f4 | f2 | f9 | f8 | f6 | f5 | f10 | f7 | 0.9752 |
| RFP | f3 | f1 | f4 | f2 | f10 | f9 | f7 | f6 | f5 | f8 | 0.9461 |

## K  THE EFFECT OF NOISE ON THE TEST ACCURACY AND FEATURE RANKING

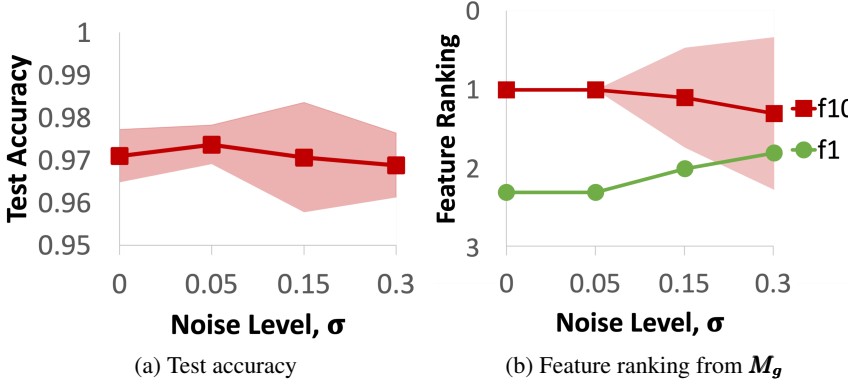

(a) Test accuracy

(b) Feature ranking from $M_g$

Figure A13: **L2X Switch dataset:** At each noise level, we ran our framework 10 times with different set of random seeds to compare how **a)** the test accuracy and **b)** the rankings of the top two features from $M_g$, i.e. $f_{10}$ and $f_1$, changes. Test accuracy improves when Gaussian noise with low variance is added. Compared to some other datasets, the feature ranking is stable with no, or low noise. Please note that we show 95% confidence interval only for feature $f_{10}$ for clarity.

## L  EXPERIMENTS ON ADULT INCOME DATASET

### L.1  ARCHITECTURE SEARCH VIA CROSS-VALIDATION AND FINAL TEST ACCURACY RESULTS

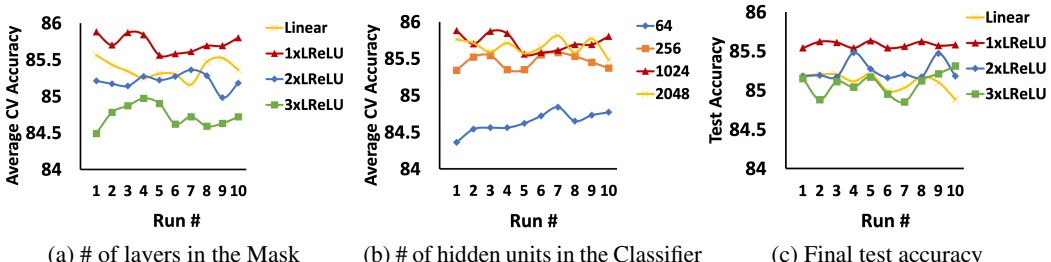

(a) # of layers in the Mask  (b) # of hidden units in the Classifier  (c) Final test accuracy

Figure A14: a) Comparing average cross validation accuracy for 10-fold cross validation (CV) by using mask generator architectures with different number of hidden layers. 10-fold CV is repeated ten times with different starting random seeds i.e. each run corresponds to one 10-fold CV run. 'Linear' means that the mask model has a single linear layer while '2xLReLU' means two hidden layers with leaky ReLU activation. In all cases, the final layer of the mask is a linear layer with sigmoid activation. b) Same experiment repeated for the 1xLReLU mask generator while increasing the number of the hidden units used in the classifier. c) The final test accuracy obtained once we settled on 1xLReLu mask and classifier with 1024 units in hidden layers and *re-run the experiments using our framework* i.e. training models on the full training set multiple times. For comparison, we included the results for other architectural choices for the mask.

We first search for the optimum number of layers for mask generator, keeping the classifier architecture fixed as [1024, 1024, 1024]. The classifer has three linear layers, two of which are followed by a leaky ReLU and dropout(p=0.2). The last layer uses sigmoid activation. We compare four choices for mask generator; i) A single linear layer with sigmoid activation, ii) A linear layer followed by leaky ReLU and another linear layer with sigmoid activation (referred as 1xLReLU) iii) two linear layers, each of which is followed by leaky ReLU, and one linear layer with sigmoid, i.e. 2xLReLU and iv) 3xLReLU. The number of hidden units in each layer in the mask generator is same as the number of features in the input (i.e. 105 in the case of Income dataset). We modified our framework to accommodate K-fold cross validation (CV). We first generated a 10-fold CV dataset from the training set. For each fold, we changed the random seed before initialising and training our models on the training fold. We obtained the validation accuracy using the corresponding validation fold. This is a slight change to our original framework, in which we train models K-times on the *same training set*. We repeated this experiment with 10-fold CV for 10 times with different set of random seeds. As shown in Figure A14a, 1xLReLU gives the best performance for all 10 repeated experiments. Please note that we also ran experiments on 1xLReLU with wider hidden layer and observed that the standard deviation in validation accuracy increases with wider mask model (not shown). Thus, we choose the number of hidden units to be same as the number of input features for all datasets and experiments throughout the paper.

We then repeated the first experiment. In this case, we used the mask generator with 1xLReLU, and varied the number of units for the hidden layers of the classifier. With everything else kept same, over-parameterised classifiers with 1024 and 2048 hidden units give the best performance. We used 1024 for the remainder of our experiments. These choices for the mask generator and the classifier are used for all other datasets and experiments since they work well as shown and discussed in the main and supplementary sections of the paper.

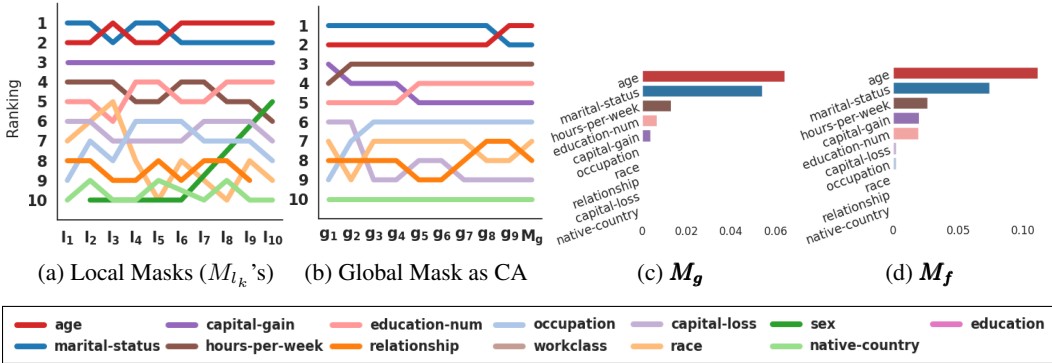

Figure A15: Using a shallow mask architecture for Income dataset. Feature importance extracted from a) Each local mask model $M_{l_k}$, referred as $l_k$ in the figure, obtained at a particular training run for 10 separate runs, b) Global model obtained by averaging the parameters of individual models up to specific run i.e. cumulative average (CA). For example, $g_3$ corresponds to averaging the parameters of first 3 local masks ($l_1, l_2, l_3$) to obtain the global mask, from which we obtain the feature importance, c) Feature importance from the global mask model obtained by averaging all 10 individual models, i.e. $M_g$ in (b), d) Feature importance from $M_f$.

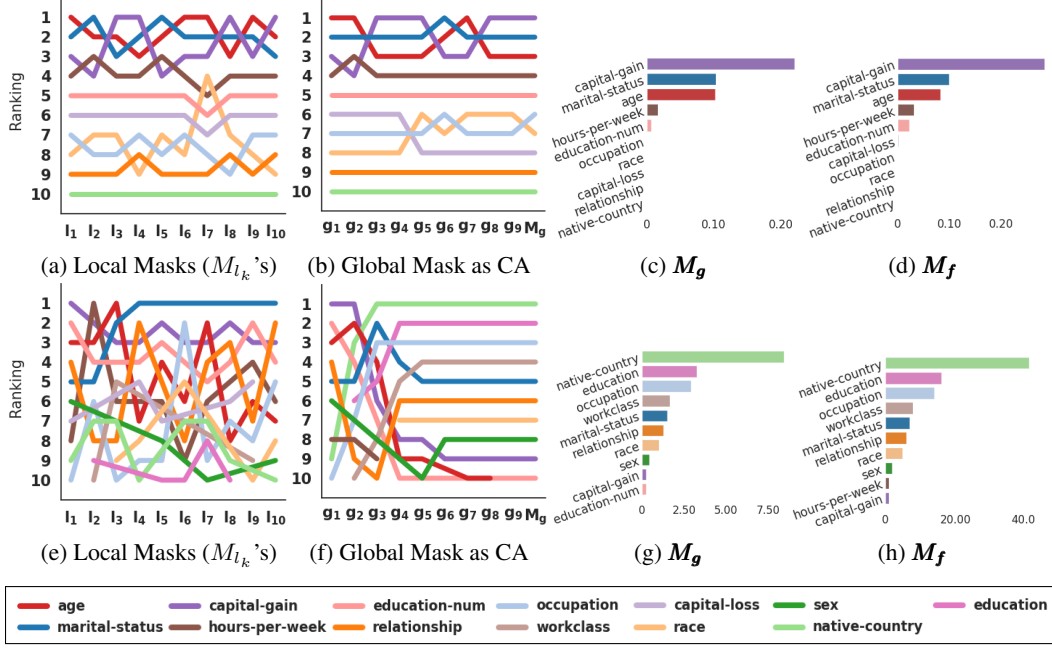

Figure A16: **First Row:** Same experiment as in Figure A15, but without noise at the input. Removing noise makes the rankings less robust. **Second Row:** Same experiment as in Figure A15, i.e. using noisy input, but the mask architecture is deeper (5 hidden layers). Parameter averaging breaks down with deeper mask model architecture (e-h).

## L.2 AVERAGING PARAMETERS OF SHALLOW NETWORKS

We start our experiments with the classification task on Income dataset to get insights into how parameter averaging works on a well-studied, real world dataset for extracting feature importance[6]. We used $p = 0.2$ when generating the binomial mask, $\beta$, and considered $\epsilon \sim \mathcal{N}(0, \sigma^2)$ with $\sigma = 0.3$.

---

[6]Unless specified otherwise, when we say feature importance, we refer to global feature importance.

As shown in Figure 1, we train our models on the whole training set for the downstream task 10 times, each time with a different random seed. We store the parameters of the trained masks, referred as local masks, from each training and denote them as $\{M_{l_1}, M_{l_2}, \ldots, M_{l_{10}}\}$. We examine the feature importance obtained from each of 10 local masks across all samples for the test set (Figure A15a). We observe that each local mask gives a slightly different ranking, especially for lower ranked features. More specifically, we could have different ranking (e.g., "age">"marital-status" vs "marital-status">"age" as top two features), depending on which seed is used when training the models. The possible reasons of this variation can be both the model initialisation as well as multicollinearity among the features of the Income dataset.

We then evaluate the effect of averaging over the parameters of the local masks on feature importance in a progressive way. To do this, we obtain a global mask $M_{g_k}$ as a cumulative average (CA) over the $k$ local masks, i.e. $M_{g_k} = 1/k \sum_{i=1}^{k} M_{l_k}$. For example, $M_{g_3}$ corresponds to averaging the parameters of the first three local masks (i.e. $M_{l_1}, M_{l_2}, M_{l_3}$ in Figure A15a). We refer to $M_{g_{10}}$ as $M_g$ for simplicity in the rest of the paper. Figure A15b shows the results for the global masks $M_{g_k}$, in which we can observe that the feature ranking becomes more stable as we use more local masks in the parameter averaging.

Figures A15c and A15d show the feature importance from $M_g = M_{g_{10}}$ and the final mask $M_f$, respectively. We first note that the weights of the features are correlated with the frequency and position of their ranks across all local masks. For example, "age" is ranked at the top more often than "marital-status" in Figure A15a, hence its weight given by $M_g$ is relatively higher than that of "marital-status". We also observe that $M_f$ moves both "capital-gain" and "capital-loss" up in the ranking.

**Effect of noise.** We further investigate the effect of removing Gaussian noise from the subsets of the masked input in Equation 2. To this end, we set $\beta = 0$, and follow the same procedure described above (Figure A15a-d). Comparing Figures A15b and A16b, we can conclude that adding noise to the input makes the global rankings more robust (a detailed comparison is in Section L.6 of the Appendix).

## L.3 ROBUSTNESS OF PARAMETER AVERAGING

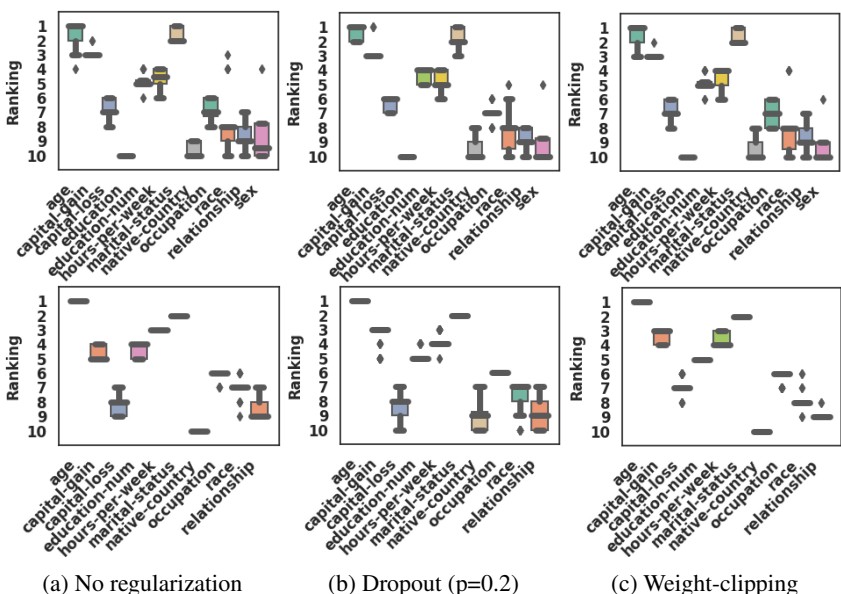

(a) No regularization      (b) Dropout (p=0.2)      (c) Weight-clipping

Figure A17: **Shallow network - Top row:** Variations in feature rankings across 20 local mask models when we apply **a)** no weight regularization, **b)** Dropout ($p = 0.2$), and **c)** Weight-clipping ($[-0.2, 0.2]$). **Bottom row**: Variations in feature rankings in 100 global models, each of which is obtained by averaging the parameters of 10 models bootstrapped from 20 local models for the same three cases.

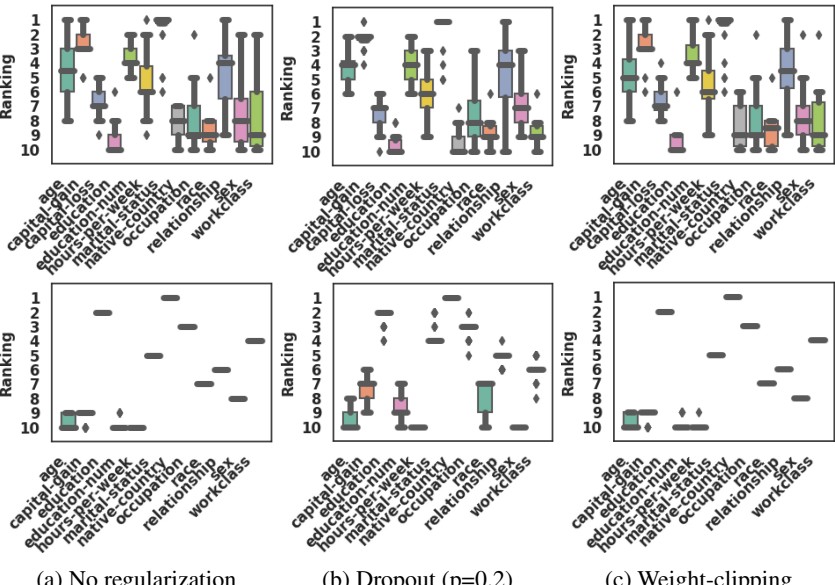

(a) No regularization      (b) Dropout (p=0.2)      (c) Weight-clipping

Figure A18: **Deep network - Top row:** Variations in feature rankings across 20 local mask models when we apply **a)** no weight regularization, **b)** Dropout ($p = 0.2$), and **c)** Weight-clipping ($[-0.2, 0.2]$). **Bottom row**: Variations in feature rankings in 100 global models, each of which is obtained by averaging the parameters of 10 models bootstrapped from 20 local models for the same three cases. Please note that the global model in the case of deeper networks results in a feature ranking that is not consistent with local model estimations, indicating that the parameter averaging does not work as well as it does for the shallow networks.

### L.4 VARIATIONS IN GLOBAL FEATURE IMPORTANCE OBTAINED FROM $M_g$ AND COMPARING IT TO THE ONES FROM TABNET(ARIK & PFISTER, 2021)

Table A6: Comparing variations in global feature importance obtained from $M_g$ across 10 different runs of our framework with different sets of random seeds to those obtained from TabNet. The rankings from TabNet are not as consistent across multiple training runs as ours. Abbreviations are; **ms:** marital status, **en:** education-num, **cg:** capital-gain, **hpw:** hours-per-week, **race:** race, **occ:** occupation, **rel:** relationship, **cl:** capital-loss, **nc:** native-country. Please note that no weight regularization is used for XTab.

| Runs | \multicolumn{10}{c}{Global Feature Importance from $M_g$ of XTab} | | | | | | | | | |
|------|-----|-----|-----|-----|-----|-----|------|-----|-----|-----|
| | 1 | 2 | 3 | 4 | 5 | 6 | 7 | 8 | 9 | 10 |
| 1 | age | ms | hpw | en | cg | occ | race | rel | cl | nc |
| 2 | age | ms | hpw | en | cg | occ | race | rel | cl | nc |
| 3 | age | ms | hpw | en | cg | occ | race | rel | cl | nc |
| 4 | age | ms | hpw | en | cg | occ | race | rel | cl | nc |
| 5 | age | ms | hpw | en | cg | occ | race | rel | cl | nc |
| 6 | age | ms | hpw | en | cg | occ | race | rel | cl | nc |
| 7 | age | ms | hpw | en | cg | occ | race | rel | cl | nc |
| 8 | age | ms | hpw | en | cg | occ | race | cl | rel | nc |
| 9 | age | ms | hpw | cg | en | race | occ | rel | cl | nc |
| 10 | age | ms | hpw | cg | en | occ | race | rel | cl | nc |

| Runs | \multicolumn{10}{c}{Global Feature Importance from TabNet} | | | | | | | | | |
|------|-----|-----|-----|-----|-----|-----|-----|------|------|------|
| | 1 | 2 | 3 | 4 | 5 | 6 | 7 | 8 | 9 | 10 |
| 1 | age | cg | en | rel | hpw | ms | sex | cl | occ | wc |
| 2 | en | ms | cg | age | cl | occ | hpw | fw | sex | race |
| 3 | en | ms | cg | sex | rel | cl | ed | age | nc | hpw |
| 4 | ms | ed | en | rel | cg | occ | cl | hpw | age | race |
| 5 | cl | ms | cg | rel | age | occ | en | hpw | wc | nc |
| 6 | ms | cg | rel | fw | en | nc | cl | race | wc | sex |
| 7 | ms | cg | en | rel | wc | occ | nc | hpw | cl | sex |
| 8 | ms | rel | occ | cg | ed | hpw | en | cl | race | nc |
| 9 | ms | cg | age | rel | occ | en | cl | hpw | race | nc |
| 10 | ms | rel | cg | wc | cl | nc | en | sex | ed | hpw |

Please note that we choose to compare Xtab and TabNet here since we can compute the rankings of the high-level categorical features in both methods. For the remaining methods, it is not easy to compute the importance score of a parent category such as "maritial-status", so we instead rank the individual categories (e.g., "single" or "married") directly as shown in Section L.5 of the Appendix.

## L.5 Comparing XTab to other methods for Adult Income dataset

Please note that, for categorical features, since it is difficult to compute the importance of a parent category from its one-hot encoded features for other methods, we compare the rankings by the individual categories (e.g., showing the importance of "single", or "married" instead of the importance of their parent category "marital-status"). Abbreviations in the tables are; **mcs:** Married civ spouse, **en:** education-num, **cg:** capital-gain, **hpw:** hours-per-week, **cl:** capital-loss, **em:** Exec-managerial, **nm:** never-married, **oc:** own-child, **os:** other-service, **fw:** Final-Weight, **mx:** Mexico, **hn:** Holand-Netherlands, **unm:** unmarried, **nm:** never-married, **phl:** Philippines, **tt:** Trinadad&Tobago, **nif:** not-in-family, **ts:** tech-support , **seni:** self-emp-not-inc.

| Runs | \multicolumn{10}{Global Feature Importance using XTab (No weight regularization is used)} | | | | | | | | | |
|---|---|---|---|---|---|---|---|---|---|---|

| **Global Feature Importance using XTab (No weight regularization is used)** | | | | | | | | | | |
|---|---|---|---|---|---|---|---|---|---|---|
| **Runs** | 1 | 2 | 3 | 4 | 5 | 6 | 7 | 8 | 9 | 10 |
| **1** | age | mcs | hpw | cg | en | em | cl | white | husband | nm |
| **2** | age | mcs | hpw | cg | en | cl | em | white | husband | nm |
| **3** | age | mcs | hpw | cg | en | em | cl | white | husband | nm |
| **4** | age | mcs | hpw | en | cg | cl | em | white | husband | nm |
| **5** | age | mcs | hpw | cg | en | cl | em | white | husband | nm |
| **6** | age | mcs | hpw | cg | en | em | cl | white | husband | nm |
| **7** | age | mcs | hpw | cg | en | cl | em | white | husband | nm |
| **8** | age | mcs | hpw | cg | en | cl | em | white | husband | nm |
| **9** | age | mcs | hpw | cg | en | cl | em | white | husband | nm |
| **10** | age | mcs | hpw | cg | en | em | cl | white | husband | nm |

| **Global Feature Importance using Saliency Maps** | | | | | | | | | | |
|---|---|---|---|---|---|---|---|---|---|---|
| **Runs** | 1 | 2 | 3 | 4 | 5 | 6 | 7 | 8 | 9 | 10 |
| **1** | mcs | cg | en | age | cl | hpw | em | ps | seni | os |
| **2** | cg | hpw | age | mcs | nm | oc | wife | en | os | cl |
| **3** | cg | age | hpw | nm | oc | mcs | wife | os | em | cl |
| **4** | cg | age | hpw | nm | oc | en | mcs | wife | os | cl |
| **5** | cg | age | hpw | mcs | oc | nm | wife | cl | en | os |
| **6** | cg | hpw | age | nm | mcs | oc | wife | cl | os | ps |
| **7** | cg | hpw | age | mcs | nm | wife | oc | os | cl | en |
| **8** | cg | age | hpw | oc | mcs | wife | husband | os | cl | em |
| **9** | cg | age | mcs | nm | os | oc | wife | en | cl | husband |
| **10** | cg | age | mcs | hpw | nm | en | oc | wife | os | cl |

| **Global Feature Importance using Integrated Gradient** | | | | | | | | | | |
|---|---|---|---|---|---|---|---|---|---|---|
| **Runs** | 1 | 2 | 3 | 4 | 5 | 6 | 7 | 8 | 9 | 10 |
| **1** | cg | mcs | age | nm | husband | hpw | en | oc | male | os |
| **2** | cg | mcs | age | nm | hpw | husband | oc | en | female | os |
| **3** | cg | mcs | age | nm | hpw | en | husband | oc | os | male |
| **4** | cg | mcs | nm | age | hpw | en | husband | oc | os | female |
| **5** | cg | mcs | age | nm | en | hpw | husband | oc | male | os |
| **6** | cg | mcs | age | hpw | nm | husband | oc | en | female | os |
| **7** | cg | mcs | age | hpw | nm | en | husband | oc | female | os |
| **8** | cg | mcs | nm | age | hpw | en | husband | oc | female | male |
| **9** | cg | mcs | nm | age | hpw | en | husband | oc | female | male |
| **10** | cg | mcs | age | nm | husband | hpw | en | oc | female | os |

| **Global Feature Importance using L2X** | | | | | | | | | | |
|---|---|---|---|---|---|---|---|---|---|---|
| **Runs** | 1 | 2 | 3 | 4 | 5 | 6 | 7 | 8 | 9 | 10 |
| **1** | cg | nm | divorced | female | mcs | en | mx | male | uk | us |
| **2** | hpw | age | divorced | nm | en | hs-grad | nif | unm | mcs | Greece |
| **3** | cg | mcs | Columbia | en | nm | oc | unm | Italy | phl | Honduras |
| **4** | cg | age | hpw | nm | em | os | fg | male | hn | Italy |
| **5** | cg | age | nm | hpw | en | South | mcs | female | Italy | phl |
| **6** | nm | mafs | cg | female | Cambodia | em | Iran | Poland | Italy | phl |
| **7** | cg | nm | oc | ts | en | male | tt | ps | mcs | hc |
| **8** | cg | os | age | nm | en | ff | divorced | af | mx | Italy |
| **9** | cg | age | nm | hpw | en | South | mcs | female | Italy | phl |
| **10** | cg | oc | age | unm | en | Doctorate | Hong | nm | hn | South |

| **Global Feature Importance using INVASE** | | | | | | | | | | |
|---|---|---|---|---|---|---|---|---|---|---|
| **Runs** | 1 | 2 | 3 | 4 | 5 | 6 | 7 | 8 | 9 | 10 |
| **1** | mcs | cg | age | en | hpw | husband | private | em | wife | cl |
| **2** | en | age | mcs | cg | hpw | husband | private | cl | fw | em |
| **3** | cg | en | age | hpw | female | wife | nif | divorced | us | cl |
| **4** | mcs | cg | en | age | hpw | em | private | male | cl | fl |
| **5** | cg | mcs | en | age | hpw | us | em | private | nif | cl |
| **6** | cg | mcs | en | age | hpw | fw | em | female | husband | bachelor |
| **7** | cg | age | en | hpw | mcs | female | us | wife | em | husband |
| **8** | cg | em | mcs | hpw | age | cl | private | husband | em | seni |
| **9** | en | cg | age | hpw | mcs | white | husband | fw | nm | private |
| **10** | mcs | em | age | cg | private | hpw | female | fw | husband | em |

### L.6 COMPARING VARIATIONS IN GLOBAL FEATURE IMPORTANCE OBTAINED FROM $M_g$ WITH AND WITHOUT NOISE AT THE INPUT

Table A7: Consistency of the ranking across multiple runs of our framework using the following two cases; Gaussian noise added to the input (top table) and the input without the noise (bottom table). Abbreviations are; **ms:** marital status, **en:** education-num, **cg:** capital-gain, **hpw:** hours-per-week, **race:** race, **occ:** occupation, **rel:** relationship, **cl:** capital-loss, **nc:** native-country

| Runs | \multicolumn{10}{l}{**Gaussian noise,** $p = 0.2$ **&** $\mathcal{N}(0, \sigma = 0.3)$**, added to the input**} |
|---|---|---|---|---|---|---|---|---|---|---|
| | 1 | 2 | 3 | 4 | 5 | 6 | 7 | 8 | 9 | 10 |
| 1 | age | ms | hpw | en | cg | occ | race | rel | cl | nc |
| 2 | age | ms | hpw | en | cg | occ | race | rel | cl | nc |
| 3 | age | ms | hpw | en | cg | occ | race | rel | cl | nc |
| 4 | age | ms | hpw | en | cg | occ | race | rel | cl | nc |
| 5 | age | ms | hpw | en | cg | occ | race | rel | cl | nc |
| 6 | age | ms | hpw | en | cg | occ | race | rel | cl | nc |
| 7 | age | ms | hpw | en | cg | occ | race | rel | cl | nc |
| 8 | age | ms | hpw | en | cg | occ | race | cl | rel | nc |
| 9 | age | ms | hpw | cg | en | race | occ | rel | cl | nc |
| 10 | age | ms | hpw | cg | en | occ | race | rel | cl | nc |

| Runs | \multicolumn{10}{l}{**No noise added to the input**} |
|---|---|---|---|---|---|---|---|---|---|---|
| | 1 | 2 | 3 | 4 | 5 | 6 | 7 | 8 | 9 | 10 |
| 1 | cg | age | ms | hpw | en | occ | race | cl | rel | nc |
| 2 | cg | ms | age | hpw | en | occ | cl | rel | race | nc |
| 3 | cg | age | ms | hpw | en | race | occ | cl | rel | nc |
| 4 | ms | age | cg | hpw | en | occ | race | rel | nc | cl |
| 5 | cg | ms | age | hpw | en | occ | race | cl | rel | nc |
| 6 | age | ms | cg | hpw | en | occ | race | rel | cl | nc |
| 7 | ms | age | cg | hpw | en | occ | race | rel | cl | nc |
| 8 | ms | age | cg | hpw | en | occ | cl | race | rel | nc |
| 9 | ms | age | cg | hpw | en | occ | cl | rel | race | nc |
| 10 | cg | ms | age | hpw | en | occ | race | rel | cl | nc |

### L.7 EXAMPLES OF INSTANCE-WISE IMPORTANCE FROM $M_f$ FOR SIX SAMPLES FROM ADULT INCOME

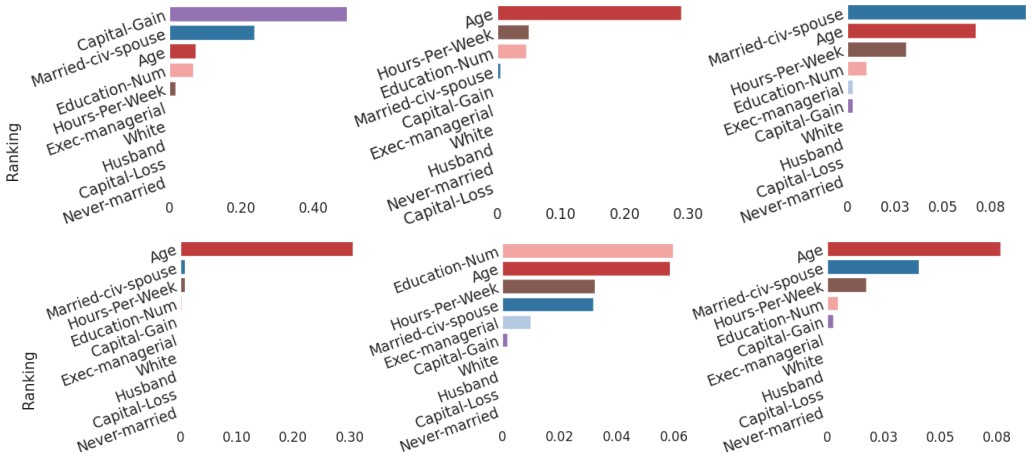

Figure A19: Examples of instance-wise feature importance for five random samples from Adult Income dataset.

L.8    SHOWING ROBUSTNESS FOR THE INSTANCE-WISE IMPORTANCE FROM $M_f$ FOR A SINGLE
SAMPLE FROM ADULT INCOME ACROSS 10 DIFFERENT EXPERIMENTS

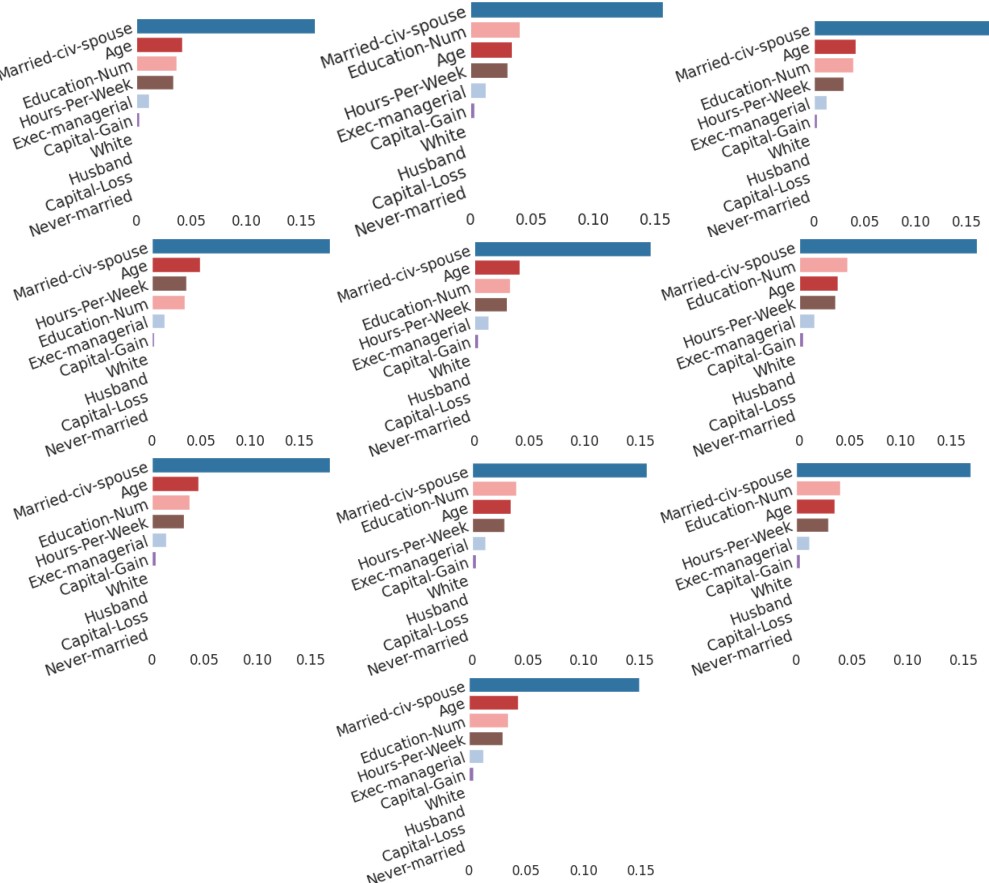

Figure A20: The robustness for the instance-wise importance from $M_f$ for a single sample from Adult Income across 10 different experiments

## M    RESULTS FOR BLOG DATASET.

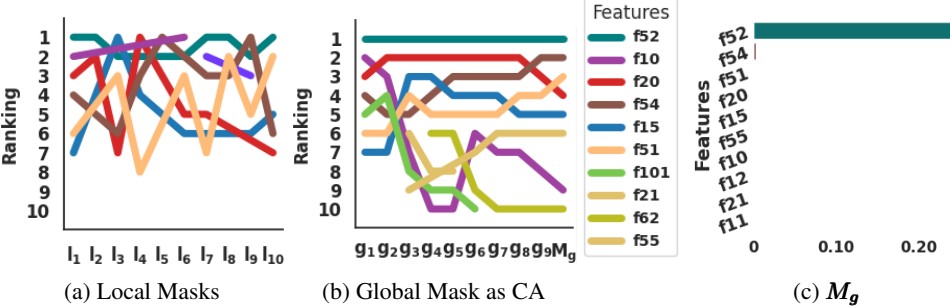

|   |   |   |
|---|---|---|
| (a) Local Masks | (b) Global Mask as CA | (c) $M_g$ |

Figure A21: **Blog dataset:** Repeating the experiment with Income dataset (i.e. using shallow network, noisy input data) in Figure A15 (a-c) for Blog dataset. We also use weight-clipping ($[-0.2, 0.2]$).

We repeated the experiment that we did for Income dataset (using shallow network and noisy input data) in Figure A15 (a-d) for Blog dataset. Looking at the Figure A21 (a-c), features **f52** (number of comments in the last 24 hours before the basetime), **f54** (number of comments in the first 24 hours after the publication of the blog post, but before basetime), **f51** (total number of comments before basetime), and **f20** (the median of **f54**) are discovered to be the most important for classifying whether a blog post would receive a comment.

### M.1    ROBUSTNESS OF PARAMETER AVERAGING

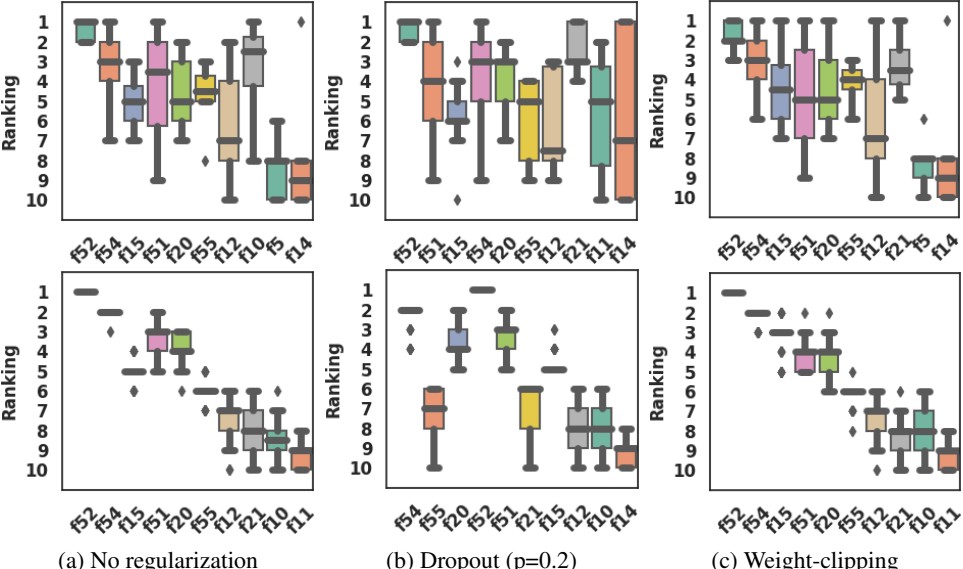

|   |   |   |
|---|---|---|
| (a) No regularization | (b) Dropout (p=0.2) | (c) Weight-clipping |

Figure A22: **Shallow network -  Top row:** Variations in feature rankings across 20 local mask models when we apply **a)** no weight regularization, **b)** Dropout ($p = 0.2$), and **c)** Weight-clipping ($[-0.2, 0.2]$). **Bottom row**: Variations in feature rankings in 100 global models, each of which is obtained by averaging the parameters of 10 models bootstrapped from 20 local models for the same three cases.

## N    OTHER WAYS OF IMPLEMENTING OUR PROPOSED METHOD

In the final training in our framework, we use $M_g$ as a reference mask while introducing another mask model to compensate for the potential sub-optimality introduced in $M_g$:

$$m_g = M_g(X) \text{ and } m_l = M_l(X) \tag{10}$$

$$m_f = (m_g + m_l)/C \text{ where } C = max(m_g + m_l) \text{ and } X_M = m_f{}^2 \odot X \tag{11}$$

However, the final mask, $m_f$, is only a scaled summation of the outputs from $M_g$ and $M_l$. A better way to do this update can be using a gating mechanism similar to input and forget gates in LSTM (Hochreiter & Schmidhuber, 1997). This would enable the model to forget the weights of some features in $m_g$ while adding more weights to others through $m_l$ in the following way:

$$m_f = \sigma(f \odot m_g + i \odot m_l) \tag{12}$$

where $\sigma$ is the sigmoid function. We leave the idea of gated masks as a future work.

## O    BROADER IMPACT

The estimation of feature ranking in many areas such as in healthcare, finance and insurance is critical in decision making process. While taking advantage of neural nets in these applications is important, we should be mindful of consistency and robustness of our methods. Our proposed method makes a contribution towards achieving a robust estimation of feature ranking. However, we should be aware of the limits and shortcomings of our approach as well as other similar approaches.

