# OpenReview forum: "Parameter Averaging for Feature Ranking"
_ICLR.cc/2023/Conference — Submitted to ICLR 2023_

### Official Review · Reviewer_iffa · 2022-10-19

**Confidence:** 4
**Correctness:** 2
**Technical Novelty And Significance:** 3
**Empirical Novelty And Significance:** 2
**Recommendation:** 5

**Clarity, Quality, Novelty And Reproducibility:**

Clarity
* Although the idea is simple, the paper is somehow very difficult to read. Some of the paper acronyms are defined in the appendix section, which is really odd. I suggest the authors to rewrite the experimental section, attempting to give a better explanation about how and against who they are comparing their algorithm.
* The architecture of the mask generator should be defined in advance. Is is also advisable to explain how the output of the mask generator is defined. If I understood it correctly, it is a sigmoid vector, which will be an advantage against higher complexity solutions like L2X. I also assume the feature importance is selected by ranking the sum of the masks of all samples. However, is it fair. In my opinion, using the mean position of the feature ranking of each sample should be better. I would like the authors to clarify why they are using the proposed solution.

Quality
* Intuitively, different NN initialization could lead to permutations in the convolutional layer channels [1], or even rotations over the dense networks [2]. As a consequence, I find the claim that the average mask generator lead to poor results is expected. However, I am intrigued about the solution provided by the authors. Does the weight averaging really helps the solution? An ablation study regarding this point should be included.

* Although the consistency of the author's solution compared to the state-of-the-art is remarkable, it does not clearly affect the results, as techniques like L2X can provide a slightly better accuracy in Table 1. Does the authors have an explanation to this phenomenon. Besides that, since they are using a synthetic algorithm, it is easy to now, for each instance, which is the feature that triggers the correct output. Thus, I suggest the authors to include an extra experiment by providing how many times the feature with the highest importance matches the one that triggers the solution.

* I would also like to see how the algorithms behave when the solution depends on a combination of multiple features. Is the algorithm capable of cope with it?

Novelty
* The idea is similar to the L2X algorithm, but introducing some modifications. The spatial complexity seems to be reduced, but the computational cost should be increased by the need of training multiple instances of the neural network.

Reproducibility
* All the information needed to reproduce the results appears to be included in the Appendix section.


[1] Ainsworth, S. K., Hayase, J., & Srinivasa, S. (2022). Git re-basin: Merging models modulo permutation symmetries. arXiv preprint arXiv:2209.04836.

[2] Weiler, M., Hamprecht, F. A., & Storath, M. (2018). Learning steerable filters for rotation equivariant cnns. In Proceedings of the IEEE Conference on Computer Vision and Pattern Recognition (pp. 849-858).

**Strength And Weaknesses:**

Strengths
* The algorithm is easy to implement.
* The consistency in the solution is remarkable.

Weaknesses
* The paper is very difficult to read and to follow. Some of the information included in the appendix should be accessible at the regular paper.
* The results are not remarkable when compared against the state-of-the-art.
* Some theoretical decisions need more clarification.

**Summary Of The Paper:**

The authors propose a novel feature ranking algorithm that consists in two parts: first, a mask generator network is used to determine the importance of each feature, in a similar way the Learning to Explain algorithm (L2X) is used. Later, the average of multiple runs (with different weight initializations) over the mask generator are obtained, which are finally introduced in a final training, combining the weights of the average mask generator with a new local one, which will establish the final ranking of each feature. The experimental results show a better consistency on how the important features are ranked, when compared with the state-of-the-art

**Summary Of The Review:**

Although I think it is an interesting paper, it is very difficult to read in its actual form. Furthermore, some decisions require an extra explanation.

---

> ### Author Response · Authors · 2022-11-11
> **Response to Reviewer iffa :  Part-1**
>
> We thank the reviewer for their response and finding our contributions significant. We should note that we revised our paper based on the feedback from all reviewers and we truly appreciate the reviewer's view on the revised paper. The following is our response to the reviewer's comments:
>
> ---
>
> - **Reviewer:**  The paper is very difficult to read and to follow. Some of the information included in the appendix should be accessible at the regular paper. Some of the paper acronyms are defined in the appendix section, which is really odd. I suggest the authors to rewrite the experimental section, attempting to give a better explanation about how and against who they are comparing their algorithm.
>
>   **Our response:**
>
>    - We thank the reviewer for the feedback and our apologies for some of the confusion in the writing. As we had many experiments to present in the paper, we shuffled around some of the sections between main paper and Appendix just before the submission, which seems to cause some of this confusion. We revised the experimental section to make it clearer as suggested. Specifically, we:
>        - Added more context in Section 2.2, clarifying the overloaded term "global".
>        - Moved the Table-1 to the Section F.1 of the Appendix, and replaced it with Figure-3a comparing various methods for the variations in their feature rankings.
>        - Added description for L2X datasets [1] in Section 3.1 and additional experiment with L2X Switch in Figure-3b.
>        - Added Figure-2d to show the performance (i.e., shortcomings) of our method for instance-wise feature importance.
>        - Revised some of the descriptions in the experimental section
>        - Revised the Method section of our paper to add more details about the mask generator.
>
> ----
>
> - **Reviewer:** The results are not remarkable when compared against the state-of-the-art. Although the consistency of the author's solution compared to the state-of-the-art is remarkable, it does not clearly affect the results, as techniques like L2X can provide a slightly better accuracy in Table 1. Does the authors have an explanation to this phenomenon.
>
>   **Our response:**
>    - We should emphasise that our paper focuses on the robustness of the feature ranking, not classification accuracy. The robustness of feature ranking comes from averaging the parameters of the mask models while the choice of classifier has an influence on classification accuracy, not the robustness.
>    - We cited test accuracy in our results just for reference. Our results show that a SOTA model such as TabNet might recommend very wrong feature ranking while giving very good test accuracy performance. So, test accuracy is by no means an indication of the correct feature ranking as we show in the paper. Also, a classical method such as Random Forest gives even a better classification accuracy than L2X [1] as shown in Table-1 of the originally submitted paper although we consider L2X as a pioneering work in the context of feature selection. So, we truly hope that the reviewer takes this perspective into account when re-scoring our work.
>    - Finally, we should note that our objective consists of two terms: cross-entropy loss + mask loss. Since we are optimising the models for both terms, there could be some tradeoff between how much the model emphasise on classification accuracy vs learning weights for the features.
>
> -----

---

> > ### Author Response · Authors · 2022-11-11
> > **Response to Reviewer iffa : Part-2**
> >
> > - **Reviewer:** -   The architecture of the mask generator should be defined in advance. Is is also advisable to explain how the output of the mask generator is defined. If I understood it correctly, it is a sigmoid vector, which will be an advantage against higher complexity solutions like L2X. I also assume the feature importance is selected by ranking the sum of the masks of all samples. However, is it fair. In my opinion, using the mean position of the feature ranking of each sample should be better. I would like the authors to clarify why they are using the proposed solution.
> >
> >   **Our response:**
> >
> >    - We revised the Method section of our paper to add more details about the mask generator. The output of the mask generator indeed uses sigmoid activation to have weights in [0,1] range.
> >
> >    - Our apologies for the confusion about the feature ranking. In the literature, there are two types of feature importance metric:
> >       - **1. The global feature importance:** It measures the average importance of a feature across all samples in a dataset (in our case, test set). In this context, the term "global" refers to how important a feature is across all samples on average.
> >       - **2. Instance-wise feature importance:** It defines the importance of a feature for a particular sample.
> >
> >     -  In this paper, we focus on Global Feature Importance. Hence, **for each feature**, we compute the average of the corresponding weight of the feature across all samples in the test set. As shown in the results, this gives a very good estimate of the global importance of a feature and we are able to rank features correctly and consistently.
> >
> >    -  Moreover, the term "global" is an overloaded term in our paper. The "global" in global feature importance has nothing to do with the "global" in global mask model. For the mask models, we just wanted to differentiate the individual models from the one obtained via parameter-averaging and we decided to use "local" vs "global" terms. Perhaps, this was a poor choice of wording, and caused some confusion.
> >
> >    -  We revised the Section 2.2 in the paper to eliminate the confusion.
> >
> > ----
> >
> > - **Reviewer:** Does the weight averaging really helps the solution? An ablation study regarding this point should be included.
> >
> >   **Our response:**
> >
> >    -  Experiments in Figure-4 and 5 in the revised paper are meant to show that the weight averaging is the one that results in the "robustness" of feature ranking. We list variations in feature ranking with and without parameter averaging (top and bottom rows respectively) under various conditions during training:
> > 	    - i) no weight regularisation,
> > 	    - ii) using dropout
> > 	    - iii) using weight-clipping.
> >
> >    -  To eliminate any doubt, we  also revised our paper by conducting an ablation study by using a simple classifier and standard training routine in Section-E of the Appendix. We repeated one of our experiments in the main paper, and show that we get the same results on SynRank dataset. Our approach is agnostic to any of the components used in our architecture. The main message of our paper is that parameter averaging of shallow neural networks results in a more robust feature ranking, and we show it over and over again through many experiments.
> >
> > ---
> >
> > - **Reviewer:**  Although the consistency of the author's solution compared to the state-of-the-art is remarkable, it does not clearly affect the results, as techniques like L2X can provide a slightly better accuracy in Table 1. Does the authors have an explanation to this phenomenon.
> >
> >   **Our response:**
> >      - As we mentioned, in the paper,  our focus is the robustness of feature ranking. We are not focusing on achieving the state of the art (SOTA) in test accuracy.  One reason why the test accuracy might be little less than L2X is that our objective includes two loss terms:
> >         - 1- Cross-entropy term for classification task
> >         - 2- Gini index to learn the mask
> >      - Since we are optimising it for both during training, there is a tradeoff. However, one could stop training the mask model and disable the Gini index term towards the end of the training to focus on classification task only. This would ensure maximizing the classification accuracy.
> > ---

---

> > > ### Author Response · Authors · 2022-11-11
> > > **Response to Reviewer iffa : Part-3**
> > >
> > > - **Reviewer:**  Since they are using a synthetic algorithm, it is easy to now, for each instance, which is the feature that triggers the correct output. Thus, I suggest the authors to include an extra experiment by providing how many times the feature with the highest importance matches the one that triggers the solution.
> > >
> > >   **Our response:**
> > >      - The reviewer refers to the performance of our method on instance-wise feature importance. We already had an experiment in the Appendix of the previously submitted paper, showing the bias of our method for instance-wise feature importance.
> > >      - We moved this experiment to the main paper  (Figure-2d in the revised paper).
> > >
> > >     ---
> > > - **Reviewer:**   I would also like to see how the algorithms behave when the solution depends on a combination of multiple features. Is the algorithm capable of cope with it?
> > >
> > >   **Our response:**
> > >
> > >      - This is a great question. Yes, this is the biggest advantage of our method. We had this experiment in the Appendix of the previously submitted paper, and moved it to the main section of the revised paper (Figure-3b)
> > >
> > >      - The experiment we are referring to is with "L2X Switch" dataset, which is originally proposed by L2X [1]
> > >      - L2X Switch dataset has 10 features. 10th feature ($f_{10}$) is used as a switch to determine whether the label is generated by the first four features ($f_1-f_4$), or the latter four ($f_5-f_8$).  9th feature ($f_9$) is not informative, and hence it is the least important feature while $f_{10}$ is the most important feature since it defines how the label should be generated. Please note that $f_{10}$'s effect on the label generation is **indirect** and there is a causal relationship between the features. Hence, all other methods fail to discover it as the most important feature consistently as shown in Figure-3b in the revised paper while our method discovers it to be the most important feature consistently. Please note that L2X algorithm [1] itself fails to discovery it completely. So, our proposed method works great in this difficult setting.
> > >      - The details of the L2X datasets can be found in Section 3.1 of the revised paper.
> > >
> > > ---
> > > - **Reviewer:**   The idea is similar to the L2X algorithm, but introducing some modifications. The spatial complexity seems to be reduced, but the computational cost should be increased by the need of training multiple instances of the neural network.
> > >
> > >   **Our response:**
> > >
> > >      - As stated in the conclusion section of our paper, when our method is used with K-fold cross-validation routine, it does not cost any extra compute. Also, in the experiments with L2X Switch dataset, we show that our method can discover the most important feature, $f_{10}$, in the setting of causally related features, while L2X fails completely. We really appreciate it if the reviewer takes this perspective into account when re-scoring our work.
> > >
> > > ---
> > >
> > > # References:
> > >
> > > [1] Jianbo Chen, Le Song, Martin Wainwright, and Michael Jordan. Learning to explain: An information theoretic perspective on model interpretation. In International Conference on Machine Learning, pp. 883–892. PMLR, 2018

---

> > > > ### Comment · Reviewer_iffa · 2022-11-27
> > > > **Concerns regarding paper summarization**
> > > >
> > > > I thank the reviewers for their extensive response. I fell the resubmitted paper added more information that wass previouly confusing. However, I think the paper cannot be properly read without the Appendix section, which is huge. The regular paper does not contain enough information to properly evaluate the impact of this work. Thus, although I would not advice against this paper to be accepted, I think this contribution should be sent to a journal paper where the page limit is not an issue.

---

> > > > > ### Author Response · Authors · 2022-11-28
> > > > > **Thanks!**
> > > > >
> > > > > We truly appreciate the reviewer's kind comments and support. We feel that the main paper has sufficient support for our claim that averaging parameters of multiple shallow networks results in a robust and accurate feature ranking. With this work:
> > > > >
> > > > >  - We hope to generate awareness about the issue of robustness (or lack thereof) in the currently available methods developed for feature ranking or selection while offering a solution for it.
> > > > >
> > > > >  - We also hope that the community will build on our work to extend it to other areas of explainable AI while addressing some of the shortcomings of our proposed method.
> > > > >
> > > > > We again thank the reviewer for the support.

---

### Official Review · Reviewer_brop · 2022-10-24

**Confidence:** 3
**Correctness:** 4
**Technical Novelty And Significance:** 2
**Empirical Novelty And Significance:** 3
**Recommendation:** 5

**Clarity, Quality, Novelty And Reproducibility:**

I feel some part of the paper could be more clear (see Minor Issues above)but overall it's of good quality.



**Strength And Weaknesses:**

Strength
- The paper motivates the problem well, and presents the idea clearly.
- Extensive numerical studies are provided to support the authors' claims.

Weaknesses
Overall I feel this paper is too "empirical" while it's not clearly supported in the paper why current design could be the best choice.
- There are several techniques borrowed from previous work (subsetting features, random noise added, etc), but without a clear justification on whether they are needed (did not seem to find in the experiments).  Naively, maybe just to compare with an average of normally trained networks without all the added tweaks?
- The authors mention that in deeper architecture the weight averaging does not work well. This raises a question on how to choose the side of the shallow networks in practice? Empirically I think there might be a sweet area for the size of shallow networks, but how do we find it and balance between the size of the network and the efficiency?

Minor Issues
- I found the method description especially the terminology on mask, classifier and encoder are a bit confusing (Maybe it's just because I am not familiar with this type of presentations).
- On the robustness vs correctness of the feature importance, I agree with the authors that it's indeed hard to know the latter but is out of the scope of this paper. There are recent work on trees related to this which are missed from the citation (e.g., https://proceedings.neurips.cc/paper/2019/hash/702cafa3bb4c9c86e4a3b6834b45aedd-Abstract.html,  https://dl.acm.org/doi/pdf/10.1145/3429445)


**Summary Of The Paper:**

The paper introduces a new method to rank feature importance in a neural network. The authors propose to train multiple shallow networks and take the average of the weights for the final ranking. The proposed method is evaluated in several datasets to prove its accuracy and stability.

**Summary Of The Review:**

The paper introduces a new method to rank feature importance in a neural network based on weights averaging from multiple shallow networks. Being a purely empirical paper, I feel some choice in the paper is not well motivated and lack supporting evidence. I believe with some revision it may be published in ICLR.

---

> ### Author Response · Authors · 2022-11-11
> **Response to Reviewer brop**
>
> We thank the reviewer for their response and finding our contributions significant. We should note that we revised our paper based on the feedback from all reviewers and we truly appreciate the reviewer's view on the revised paper. The following is our response to the reviewer's comments:
>
> ---
>
> - **Reviewer:** Overall I feel this paper is too "empirical" while it's not clearly supported in the paper why current design could be the best choice... There are several techniques borrowed from previous work (subsetting features, random noise added, etc), but without a clear justification on whether they are needed (did not seem to find in the experiments). Naively, maybe just to compare with an average of normally trained networks without all the added tweaks?
>
>   **Our response:**
>   - Our approach is agnostic to any of the components used in our architecture. In the paper, we adopted SubTab framework [1] because we wanted to extend our work to pre-trained models in the future and this was a good opportunity to lay the groundwork for future follow-up works. Perhaps, this choice was a mistake since it seems to cause some confusion.
>   - To eliminate any doubt, we revised our paper by conducting an ablation study by using a simple classifier and standard training routine in Section-E of the Appendix of the revised paper. We repeated one of our experiments in the main paper, and show that we get the same results on SynRank dataset. The main message of our paper is that parameter averaging of shallow neural networks results in a more robust feature ranking, and we show it over and over again through many experiments. We truly hope that the reviewer keeps this perspective in mind when re-evaluating our revised paper.
>   - We should emphasise that our paper focuses on the robustness of the feature ranking, not classification accuracy. The robustness of feature ranking comes from averaging the parameters of the mask models. The choice of Encoder/Classifier has an influence on classification accuracy, not the robustness.
>
> ---
> - **Reviewer:** The authors mention that in deeper architecture the weight averaging does not work well. This raises a question on how to choose the side of the shallow networks in practice? Empirically I think there might be a sweet area for the size of shallow networks, but how do we find it and balance between the size of the network and the efficiency?
>
>   **Our response:**
>   - In the paper, we choose the number of units in the shallow network (its width) to be same as the number of features of the raw input data. This is ideal since we just want to compute a weight per corresponding feature. Wider shallow network can be used, but it would just be more redundant than what is needed. As a side note, we experimented with wider mask models and observed that it could result in more variations in feature ranking.
>
> ---
>
> - **Reviewer:** I found the method description especially the terminology on mask, classifier and encoder are a bit confusing (Maybe it's just because I am not familiar with this type of presentations).
>
>   **Our response:**
>   - Our apologies for the confusion. As we mentioned before, we adopted SubTab framework [1], which proposes to learn representations by using subsets of features. We thought that  we could extend our work to pre-trained models in the future and that we wanted to lay the groundwork for future follow-up works. In retrospect, this choice was a mistake since it seems to cause some confusion. However, in our ablation study in **Section E** of the Appendix in the revised paper, we show that the main claim of the paper ( i.e. averaging parameters of shallow networks leads to more robust feature ranking) is still true regardless of the architecture that comes after the mask model.
>   - If there are other areas of the paper that needs some clarification, we would be more than happy to revise the relevant sections of the paper.
>
> ---
>
> - **Reviewer:** On the robustness vs correctness of the feature importance, I agree with the authors that it's indeed hard to know the latter but is out of the scope of this paper. There are recent work on trees related to this which are missed from the citation.
>
>   **Our response:**
>
>   - Many thanks for the references. We added them as additional citations in our revised paper.
>
>
> # References:
>
> [1] Talip Ucar, Ehsan Hajiramezanali, and Lindsay Edwards. Subtab: Subsetting features of tabular data for self-supervised representation learning. Advances in Neural Information Processing Systems, 34, 2021.

---

### Official Review · Reviewer_ZHfL · 2022-10-24

**Confidence:** 3
**Correctness:** 2
**Technical Novelty And Significance:** 2
**Empirical Novelty And Significance:** 2
**Recommendation:** 3

**Clarity, Quality, Novelty And Reproducibility:**

I found parts of the paper difficult to read. Authors refer to definitions such as generate_subsets(X_M) which are important but not defined the main paper and one has to hunt through the supplementary material to find them. I think the main paper should be as self contained as possible instead of constantly referring to the supplementary. Also, the full submission is 37 pages long which is excessive for a 9 page conference paper, if such length is needed I would consider a longer journal submission.

**Strength And Weaknesses:**

I found the proposed approach interesting but very ad hoc. There are quite a few components such as feature masks (local and global), feature subsets, feature noise, encoder that generates an embedding for each feature subset and finally a classifier. It is unclear which of these components are needed to improve feature importance stability and which are for classification performance. It is also unclear what authors mean by "global" feature ranking. In my understanding feature importance is typically model specific. Tabular datasets often contain subsets of highly correlated features and typically multiple such subsets can be used to get accurate models. Artifacts such as random seed influence which subset (or subsets) the model converges to during training. This is commonly observed in xgboost where, when optimized under the "hist" setting and random seed, the trees tend to pick a subset of the correlated features. Averaging together models is likely going to spread importance over correlated features and I don't quite see how this "global" importance is useful.

**Summary Of The Paper:**

The paper proposes an approach to estimate feature importance. Multiple shallow mask models are trained with different random seeds. These resulting weights and then averaged to produce a global model that can be inferior in accuracy but better identify the global feature importance.

**Summary Of The Review:**

I think the proposed approach is interesting but the paper needs a considerable revision to both justify the goal and each of the proposed components including a detailed ablation study.

---

> ### Author Response · Authors · 2022-11-11
> **Response to Reviewer ZHfL**
>
> We thank the reviewer for their response and finding our work interesting. We should note that we revised our paper based on the feedback from all reviewers and we truly appreciate the reviewer's view on the revised paper. The following is our response to the reviewer's comments:
>
> - **Reviewer:** The reviewer finds the proposed approach interesting but very ad hoc since it is unclear to the reviewer that which of the components are needed to improve feature importance stability and which are for classification performance.
>
>   **Our response:**
>
>   - We should emphasise that our paper focuses on the robustness of the feature ranking, not classification accuracy. The robustness of feature ranking comes from averaging the parameters of the mask models. The choice of Encoder/Classifier has an influence on classification accuracy, not the robustness.
>
>   - Our approach is agnostic to any of the components used in our architecture. In the paper, we adopted SubTab framework [1] because we wanted to extend our work to pre-trained models in the future and this was a good opportunity to lay the groundwork for future follow-up works.
>
>   - To eliminate any doubt, we revised our paper by conducting an ablation study by using a simple classifier and standard training routine in Section-E of the Appendix. We repeated one of our experiments in the main paper, and show that we get the same results on SynRank dataset. The main message of our paper is that **parameter averaging of shallow neural networks results in a more robust feature ranking**, and we show it over and over again through many experiments.
>
> ------
>
> - **Reviewer:** It is also unclear what authors mean by "global" feature ranking. ....Averaging together models is likely going to spread importance over correlated features and I don't quite see how this "global" importance is useful.
>
>   **Our response:**
>   - Our apologies for the confusion. The term "global" is an overloaded term in our paper, and we revised the Section 2.2 in our revised paper to eliminate the confusion.
>   - In the literature, there are two types of feature importance metric:
>        - **1. The global feature importance:** It measures the average importance of a feature across all samples in a dataset (in our case, test set). In this context, the term "global" refers to how important a feature is across all samples on average.
>        - **2. Instance-wise feature importance**: It defines the importance of a feature for a particular sample.
>   - Thus, "global" in global feature importance has nothing to do with the "global" in global mask model. For the mask models, we just wanted to differentiate the individual models from the one obtained via parameter-averaging and we decided to use "local" vs "global" terms. Perhaps, this was a poor choice of wording, and caused some confusion.
>
> ------
>
> - **Reviewer:** I found parts of the paper difficult to read. Authors refer to definitions such as generate_subsets(X_M) which are important but not defined the main paper and one has to hunt through the supplementary material to find them.
>
>   **Our response:**
>   - We thank the reviewer for this feedback. We revised our paper to make parts of it more clear. We appreciate any further feedback if any particular parts of the paper needs more improvement.
>
> ------
>
> - **Reviewer:** The full submission is 37 pages long which is excessive for a 9 page conference paper, if such length is needed I would consider a longer journal submission.
>
>   **Our response:**
>   - Our main paper is a 9 page conference paper. The appendix is optional, and provides a list of repeated experiments using different datasets as well as some more details. However, the main paper is sufficient to support our claims. Moreover, since the paper is an empirical study, we feel that we have to run extensive experiments to show the validity of our approach. Not doing so would be more worrisome. Also, the papers with long appendix are not uncommon these days. We truly hope that the reviewer takes this perspective into account when re-scoring our work.
>
> **References:**
>
> [1] Talip Ucar, Ehsan Hajiramezanali, and Lindsay Edwards. Subtab: Subsetting features of tabular data for self-supervised representation learning. Advances in Neural Information Processing Systems, 34, 2021.

---

### Official Review · Reviewer_G9vP · 2022-10-25

**Confidence:** 3
**Clarity, Quality, Novelty And Reproducibility:** The proposed method is novel, but no …
**Correctness:** 3
**Technical Novelty And Significance:** 3
**Empirical Novelty And Significance:** 3
**Recommendation:** 5

**Strength And Weaknesses:**

Strength: Experimental results are good.
Weaknesses: There is no experimental validation on real task.

**Summary Of The Paper:**

This paper introduces a novel method based on parameter averaging to estimate accurate and robust feature importance in tabular data setting. The proposed method first initializes and trains multiple instances of a shallow network (referred as local masks) with different random seeds for a downstream task and then obtains a global mask model by averaging the parameters of local masks. The experimental results show that although the parameter averaging might result in a global model with higher loss, it still leads to the discovery of the ground-truth feature importance more consistently than an individual model does.

**Summary Of The Review:**

The proposed method is novel, but no practical application validation.

---

> ### Author Response · Authors · 2022-11-11
> **Response to Reviewer G9vP**
>
> We thank the reviewer for their response and for finding our method novel. We should note that we revised our paper based on the feedback from all reviewers and we truly appreciate the reviewer's view on the revised paper. The following is our response to the reviewer's comments:
>
> - **Reviewer:** There is no practical application validation in the paper.
>
>   **Our response:**
>
>     - Indeed, this is a very common criticism addressed to the works in tabular domain. The lack of good benchmark datasets is one of the major obstacles in deep learning research in tabular domain. Hence, many pioneering works [1, 2] turn to synthetic datasets to validate their method, and we followed this common practice to validate ours as well. In fact, we used the same synthetic datasets proposed and used by [1, 2]  to compare our method to theirs for the robustness of feature ranking.
>
>     - Moreover, the aforementioned pioneering works also received similar feedbacks during their reviews. Validation using real world datasets will be an ongoing issue in the tabular domain. We truly hope that the reviewer keeps this perspective in mind when re-scoring our work. We are also open to validating our approach on any dataset that the reviewer might have in mind.
>
> ------
>
> - **Reviewer:** Some of the paper’s claims have minor issues. A few statements are not well-supported, or require small changes to be made correct.
>
>   **Our response:**
>
>     - We made some changes to the paper and added some clarifications in parts of the paper. We also moved some sections from the appendix to the main paper to give a more compressive view of our work. For example, to show the advantage of our method, we moved our experiments with L2X Switch dataset (a dataset proposed by [1], to the main paper. L2X Switch dataset provides the hardest task in feature ranking since the most important feature, $f_{10}$,  influences the label generation indirectly by determining which other features to be used when generating the class label (i.e. there is a causal relationship between the features, in which $f_{10}$ acts as a parent feature). All other models fail to discover $f_{10}$ as the most important feature while our method ranks it as the top feature consistently as shown in Figure-3b in the revised paper.
>     - Moreover, we would really appreciate it if the reviewer can point out what other points might not be well supported, or might have minor issues in the paper. We are open to addressing them during the rebuttal process.
>
>
> **References:**
>
> [1] Jianbo Chen, Le Song, Martin Wainwright, and Michael Jordan. Learning to explain: An information theoretic perspective on model interpretation. In International Conference on Machine Learning, pages 883–892. PMLR, 2018
>
> [2] Jinsung Yoon, James Jordon, and Mihaela van der Schaar. Invase: Instance-wise variable selection using neural networks. In International Conference on Learning Representations, 2018.

---

### Decision · Program_Chairs · 2023-01-20

**Decision:**

Reject

**Justification For Why Not Higher Score:**

The submission has consistent concerns from the reviewers on whether the method is ad hoc (choice of base models, etc.), evaluation, and self contained readability.  The reviewers were unanimous in their scores being below the threshold for acceptance.

**Justification For Why Not Lower Score:**

N/A

**Metareview: Summary, Strengths And Weaknesses:**

The submission proposes a strategy of training multiple shallow networks and performing parameter averaging to assess feature importance.  The main findings are that parameter averaging results in higher loss, but more stable feature detection, both of which are plausible outcomes from such a scheme.  The reviewers were unanimous that the submission falls below the threshold for acceptance to ICLR.  There were (debated) critiques about the form of evaluation, with concerns from reviewers about the reliance on synthetic evaluation and the response from the authors that the synthetic evaluations follow those of high profile papers from 2018 (https://openreview.net/forum?id=NYtq-CsRP3H&noteId=lHiY0M8IWC).  However, a clear and consistent issue raised by multiple reviewers is that the paper is not self contained without reference to the very long appendix section.  This was debated by the authors, but the concerns remain post rebuttal, and multiple reviewers suggest that a more appropriate publication strategy for the work in its current form would be a journal submission with an integrated longer presentation - though I would imagine concerns about e.g. synthetic evaluation, whether the method is too ad hoc, ablation study, etc., might be of concerns to journal reviewers as well.